# Dynamic Diameter in High-Dimensions against Adaptive Adversary and Beyond[*]

**Kiarash Banihashem**[†]
University of Maryland
College Park, MD, USA
kiarash@umd.edu

**Jeff Giliberti**[†]
University of Maryland
College Park, MD, USA
jeffgili@umd.edu

**Samira Goudarzi**[†]
University of Maryland
College Park, MD, USA
samirag@umd.edu

**MohammadTaghi Hajiaghayi**[†]
University of Maryland
College Park, MD, USA
hajiagha@umd.edu

**Peyman Jabbarzade**[†]
University of Maryland
College Park, MD, USA
peymanj@umd.edu

**Morteza Monemizadeh**[†]
TU Eindhoven
Eindhoven, The Netherlands
M.Monemizadeh@tue.nl

## Abstract

In this paper, we study the fundamental problems of maintaining the diameter and a $k$-center clustering of a dynamic point set $P \subset \mathbb{R}^d$, where points may be inserted or deleted over time and the ambient dimension $d$ is not constant and may be high. Our focus is on designing algorithms that remain effective even in the presence of an *adaptive adversary*—an adversary that, at any time $t$, knows the entire history of the algorithm's outputs as well as all the random bits used by the algorithm up to that point. We present a fully dynamic algorithm that maintains a 2-approximate diameter with a *worst-case* update time of $\mathrm{poly}(d, \log n)$, where $n$ is the length of the stream. Our result is achieved by identifying a robust representative of the dataset that requires infrequent updates, combined with a careful deamortization. To the best of our knowledge, this is the first efficient fully-dynamic algorithm for diameter in high dimensions that *simultaneously* achieves a 2-approximation guarantee and robustness against an adaptive adversary. We also give an improved dynamic $(4 + \epsilon)$-approximation algorithm for the $k$-center problem, also resilient to an adaptive adversary. Our clustering algorithm achieves an amortized update time of $k^{2.5}d \cdot \mathrm{poly}(\epsilon^{-1}, \log n)$, improving upon the amortized update time of $k^6 d \cdot \mathrm{poly}(\epsilon^{-1}, \log n)$ by Biabani et al. [NeurIPS'24].

## 1 Introduction

Maintaining representative properties of a dynamic point set, such as its current diameter and $k$-center clusters, is a fundamental computing task. Given a set of points $P$ in a $d$-dimensional Euclidean space, with $P$ subject to insertions and deletions, our goal is to efficiently compute and maintain approximate diameter, minimum enclosing ball and $k$-center clustering. We are interested in algorithms that are robust against an adaptive adversary, that is, the algorithm performance is evaluated against an instance that adapts to the previous choices of the algorithm.

Approximating the *diameter* of a dataset—the greatest pairwise distance—is a fundamental geometric operation with direct relevance to machine learning. It is commonly used in clustering to measure

---

[*]The work is partially supported by DARPA QuICC, ONR MURI 2024 award on Algorithms, Learning, and Game Theory, Army-Research Laboratory (ARL) grant W911NF2410052, NSF AF:Small grants 2218678, 2114269, 2347322.

[†]Equal Contribution

39th Conference on Neural Information Processing Systems (NeurIPS 2025).

the spread of a cluster, where a smaller diameter indicates that data points within a cluster are more similar to each other. In hierarchical clustering algorithms, diameter thresholds are often used to determine whether clusters should be merged or split [Liu et al., 2012]. In outlier detection, points that substantially increase the diameter of a dataset can be flagged as anomalies [Aggarwal, 2016]. Diameter also arises in approximate furthest neighbor queries [Pagh et al., 2015], and in active learning, where it quantifies the uncertainty over hypotheses [Tosh and Dasgupta, 2017]. These applications become particularly challenging in high-dimensional settings, where exact diameter computation is expensive and geometric properties often degrade [Indyk, 2000, Goel et al., 2001], making standard distance-based methods less effective and harder to analyze [Indyk and Motwani, 1998, Har-Peled et al., 2012].

*Minimum Enclosing Ball (MEB)* is closely related to diameter and plays a central role in high-dimensional data analysis, particularly in one-class SVMs [Tax and Duin, 2004] and Support Vector Data Description (SVDD) [Frandi et al., 2010]. A MEB captures the spread of the data by enclosing it in the smallest possible hypersphere, whose center and radius define the classifier's parameters and decision boundary. In dual SVM formulations, solving the MEB corresponds to identifying support vectors that lie on the boundary of the ball. Fast and efficient MEB computation is thus crucial for scaling SVM training to large, high-dimensional datasets [Badoiu and Clarkson, 2003, Clarkson, 2010, Clarkson et al., 2012], where the cost of exact geometric operations can become a bottleneck.

**Diameter Problem.** Designing dynamic algorithms for approximating the diameter of a point set in Euclidean space is particularly challenging when facing an *adaptive adversary*—one that reacts to the algorithm's internal randomness or past outputs. A folklore (dynamic) 2-approximation algorithm (e.g., Chan and Pathak [2014]) picks a random anchor point $p \in P$ and maintains its furthest neighbor throughout updates. This strategy has $O(d \log n)$ amortized update time, since the anchor is deleted with probability $1/n$ for $n = |P|$.

However, the algorithm is fragile: if the adversary deletes $p$, the guarantee fails, and the procedure must be restarted—potentially requiring $\Omega(n)$ time. Surprisingly, the update time of this is simple and vulnerable method remains the best known when faced with an adaptive adversary. Developing a robust, fully dynamic algorithm with any non-trivial approximation that has worst-case $\text{poly}(d \log n)$ update time remains an important open problem. We summarize existing bounds in Table 1.

| Approximation | Update Time | Guarantee | Adversary | Algorithm |
|:---:|:---:|:---:|:---:|:---:|
| 2 | $O(d \log n)$ | Amortized | Oblivious | Folklore |
| $1 + \epsilon$ | $O(d \cdot n^{1/(1+\epsilon)^2})$ | Worst-case | Oblivious | [Indyk, 2003] |
| $1 + \epsilon$ | $\epsilon^{-\Theta(d)}$ | Worst-case | Deterministic | [Zarrabi-Zadeh, 2011] |
| $\sqrt{2} + \epsilon$ | $O(d \log n)$ | Amortized | Oblivious | [Chan and Pathak, 2014] |
| 2 | $O(d^5 \log^3 n)$ | Worst-case | Adaptive | **This paper** |

Table 1: State-of-the-art algorithms for fully-dynamic diameter.

$k$**-Center Clustering.** The $k$-center problem seeks $k$ representative points that minimize the maximum distance to any point in the dataset, and is a well-studied objective in clustering. In the static setting, a simple greedy algorithm achieves a 2-approximation [Gonzalez, 1985, Hochbaum and Shmoys, 1986], which is NP-hard to improve for any constant factor [Hsu and Nemhauser, 1979]; in Euclidean spaces, this can be refined to 1.822 [Feder and Greene, 1988, Bern and Eppstein, 1997]. Dynamic variants of $k$-center clustering have attracted significant interest (e.g., Charikar et al. [1997], Chan et al. [2018], Bateni et al. [2023], Łącki et al. [2024]), with a focus on balancing approximation quality, update time, and robustness. Similar to the diameter problem, key performance measures include: (1) the approximation ratio; (2) per-update time (amortized or worst-case); and (3) resilience to different adversary models (oblivious, adaptive, or deterministic). Table 2 summarizes the state-of-the-art across these dimensions; see Appendix A for additional discussion.

Despite recent advances, the complexity of fully dynamic $k$-center clustering against an adaptive adversary remains poorly understood. In particular, there is a significant gap between the nearly optimal approximation and update time guarantees in the oblivious setting [Bateni et al., 2023] and the recent $(4 + \varepsilon)$-approximation by [Biabani et al., 2024], which incurs an amortized update time of $\tilde{O}(k^6)$—prohibitively large even for small $k$. Sacrificing update time further, [Goranci et al., 2021] achieve a $2 + \varepsilon$ approximation with update time exponential in $d$, which is practical only for small

| Approximation | Update Time | Guarantee | Adversary | Algorithm |
|:---:|:---:|:---:|:---:|:---:|
| $2 + \epsilon$ | $2^{O(d)} \cdot \widetilde{O}(1)$ | Worst-case | Deterministic | [Goranci et al., 2021] |
| $2 + \epsilon$ | $\widetilde{O}(k/\epsilon)$ | Amortized | Oblivious | [Bateni et al., 2023] |
| $O(\min\{k, \frac{\log(n/k)}{\log\log n}\})$ | $\widetilde{O}(k)$ | Amortized | Deterministic | [Bateni et al., 2023] |
| $O(1)$ | $\text{poly}(n, k)$ | Worst-case | Deterministic | [Łącki et al., 2024] |
| $4 + \epsilon$ | $\widetilde{O}(k^6)$ | Amortized | Adaptive | [Biabani et al., 2024] |
| $4 + \epsilon$ | $\widetilde{\mathbf{O}}(\mathbf{k^{2.5}})$ | Amortized | Adaptive | **This paper** |

Table 2: State-of-the-art algorithms for fully dynamic $k$-center clustering. The $\tilde{O}(\cdot)$ notation hides $d$ and $\text{poly} \log(k, n, \rho, \epsilon)$ factors, where $\rho$ denotes the spread ratio of the dataset. Our algorithm is for Euclidean spaces, while the other results are for general metric spaces. The algorithm of [Biabani et al., 2024] is developed for $k$-center with outliers.

dimensions. More recently, Łącki et al. [2024], Forster and Skarlatos [2025] obtained deterministic $O(1)$-approximation algorithms with polynomial update time in $n$.

**Adaptive adversary.** Dynamic algorithms [Onak and Rubinfeld, 2010, Baswana et al., 2018, Abraham et al., 2016] have commonly assumed an *oblivious adversary*, that is, the adversary must choose in advance an instance against which the algorithm is evaluated. Unfortunately, it turns out that the assumption of an oblivious adversary is problematic in many natural scenarios. This is the case when a dynamic algorithm is used as a subroutine inside another algorithm, when a database update depends on a previous one, or when the algorithm is faced with adversarial attacks.

In theoretical computer science and machine learning, several notions of *adaptive adversary* have been studied extensively. We consider an adaptive adversary who has full knowledge of the algorithm and can see its random choices up to that point, but not future ones. This is a natural and a central notion of adversary: it is the strongest possible adversary against which a randomized algorithm can be evaluated. The adversary is able to decide future updates based on the past decisions of the algorithm, and its updates might force the algorithm to misbehave, by either producing an incorrect answer or taking a long time to run. This definition of adversary closely reflects the *white-box adversarial model* in robust machine learning [Ilyas et al., 2019, Madry et al., 2018].

In the white-box setting, an adversary has access to the internal structure of a machine learning model, including its parameters and training data, such as its parameters or training weights. This level of access makes the adversary particularly powerful: recent work has shown that even subtle input perturbations can mislead the model into making incorrect predictions (see [Biggio et al., 2013, Athalye et al., 2018] and references therein). As a result, developing algorithms that are robust to white-box adversarial attacks has become a central focus in robust machine learning [Ilyas et al., 2019, Madry et al., 2018, Tramèr et al., 2018, Kurakin et al., 2017, Liu et al., 2017], as well as in dynamic [Nanongkai and Saranurak, 2017, Cherapanamjeri and Nelson, 2020, Wajc, 2020], streaming [Ben-Eliezer and Yogev, 2020, Ajtai et al., 2022], and other BigData settings [Mironov et al., 2008, Bogunovic et al., 2017].

The robustness of fully-dynamic algorithms against adaptive adversaries has recently received significant attention [Nanongkai and Saranurak, 2017, Wajc, 2020, Bateni et al., 2023, Roghani et al., 2022]. Similar notions have also been studied in the streaming model [Ajtai et al., 2022, Feng and Woodruff, 2023, Feng et al., 2024]. We emphasize that our notion of adaptive adversary is significantly stronger than the commonly studied model, where the adversary can observe the outputs of the algorithm over time but not its internal random choices. To distinguish the two, we refer to our stronger model as the *randomness-adaptive adversary* and the weaker one as the *output-adaptive adversary*. This distinction parallels the difference between white-box and black-box adversarial attacks in machine learning.

A key distinction is that, under an output-adaptive adversary, one can often design algorithms that carefully manage dependencies between internal randomness and observable outputs, thus preventing information leakage. Such techniques, however, break apart when the adversary has visibility into the random choices themselves. For many problems—including diameter and $k$-center clustering—the gap between output-adaptive and randomness-adaptive adversaries remains poorly understood.

## 1.1 Our Contribution

We design a fully-dynamic algorithm for approximating the diameter in high-dimensional Euclidean spaces that is robust to an adaptive adversary. The same algorithm extends to maintaining a minimum enclosing ball (MEB) that encloses the given point set. Our main result is the following:

**Theorem 1.1.** *For the diameter and minimum enclosing ball (or $1$-center) problems in $d$ dimensions, there exists a fully-dynamic algorithm that achieves a $2$-approximation with success probability at least $1 - \delta$. The algorithm works against a randomness-adaptive adversary and guarantees a worst-case update time of $O\left(d^5 \log^{1.5}(d) \log^{1.5}(n/\delta)\right)$, where $n$ is the length of the update stream.*

Our algorithm is the first to be resilient against a randomness-adaptive adversary, while achieving a non-trivial constant approximation in high dimensions. Previously, efficient algorithms were known only for small values of $d$ [Zarrabi-Zadeh, 2011].

We extend some techniques developed for the diameter problem to the $k$-center clustering problem for $k \geq 2$, resulting in the following result that improves the amortized update time $k^6 \cdot \text{poly}(d, \varepsilon^{-1}, \log n)$ of [Biabani et al., 2024].

**Theorem 1.2.** *Let $k \in \mathbb{N}$ and $0 < \varepsilon \leq 1$. For the $k$-center problem in $d$ dimensions, there exists a fully-dynamic algorithm that achieves a $(4 + \varepsilon)$-approximation with success probability at least $1 - \delta$. The algorithm works against a randomness-adaptive adversary and guarantees an amortized update time of $k^{2.5}d \cdot \text{poly}(\log n, \varepsilon^{-1}, \delta)$, where $n$ is the length of the update stream.*

**Overview of our techniques.** One of our main technical contributions is the development of algorithms that remain resilient to adversarial updates, even when the adversary has full knowledge of the current state of the algorithm. To withstand such an adversary, our algorithms cannot rely on random decisions; instead, their decisions must be provably robust. This requirement rules out many commonly used techniques in dynamic algorithms for achieving fast update times or black-box robustness. In particular, it eliminates the use of random decisions Chan and Pathak [2014] (e.g., an adversary deletes a randomly chosen point from $P$ with probability $1/|P|$), as well as the use of differential privacy [Cherapanamjeri et al., 2023, Hassidim et al., 2022] that includes reporting noisy answers and obfuscating the state of the algorithm through multiple copies with varying randomness.

Although our dynamic algorithms are randomized, randomness is used solely to accelerate the decision-making process. This contrasts fundamentally with prior dynamic algorithms against *oblivious adversaries* [Indyk, 2003, Chan and Pathak, 2014, Bateni et al., 2023], where randomness is typically used to make decisions that adversaries cannot exploit.

For both the diameter and $k$-center problems, we quantify the robustness of a "decision" by the number of updates it can withstand. Intuitively, a decision requiring $O(T)$ time to compute is considered $\epsilon$-robust if it remains valid after applying $O(\epsilon T)$ changes to the underlying dataset, for some $0 < \epsilon < 1$. Beyond $O(\epsilon T)$ updates, the data may have changed significantly, making even a robust decision irrelevant. Thus, the algorithm must re-compute and establish a new "robust" decision capable of withstanding the subsequent updates. Our dynamic diameter and k-center algorithms both rely on this iterative approach.

In the case of the diameter problem, we compute an $\varepsilon$-*robust representative point* (Definition 2.2) that continues to be representative of the surviving point set (i.e., points that are inserted but not deleted in the dynamic setting) even after $O(\varepsilon T)$ deletions. However, it is not immediately obvious whether such $\varepsilon$-robust representatives exist or whether they can be computed efficiently. The core novelty of our framework lies in showing that such points can indeed be constructed and maintained efficiently. The construction step relies on the notion of *centerpoint* [Har-Peled and Jones, 2020, Clarkson et al., 1993] (see Definition 2.5). Informally, a centerpoint is a point that lies *"deep"* within a point set. To our knowledge, this is the first application of centerpoints in dynamic algorithms, and we believe this technique has broader potential in developing dynamic and streaming algorithms resilient to adaptive adversaries.

For the $k$-center problem, the robustness of a cluster center $c$ is quantified by the number of nearby points—those within some distance $D$. Since any point lies within distance at most $OPT$ of an optimal center $c^*$, a previously opened center $c$ that still retains a nearby point (within distance $D$) can cluster all points assigned to $c^*$ using radius $D + 2OPT$. This idea is also used in prior work on

$k$-center clustering [Biabani et al., 2024, Cohen-Addad et al., 2016]. Our main contribution for this problem is a faster update time through a more careful random sampling, which we overview below.

To find robust centers, we sample $\widetilde{O}(k)$ points and, for each sampled point, measure the fraction of other sampled points that lie within a ball of radius $2 \cdot OPT$ around it. This identifies a sampled point that is robust with respect to the sample in total time $\widetilde{O}(k^2)$. This procedure is efficient when the number of clusters is relatively small. If $k \leq n^{2/3}$ the above sampling and pairwise coverage tests suffice, giving an overall cost of $\widetilde{O}(k^2)$.

When $k > n^{2/3}$, we reduce the sample size to $\widetilde{O}(\sqrt{k})$ and change the robustness test: rather than testing coverage among sampled points, we test for each sampled point the fraction of *original* points in $P$ that lie within radius $2 \cdot OPT$, incurring an overall cost of $\widetilde{O}(\sqrt{k}\, n)$. Two scenarios now cover all possibilities. First, if the largest $\sqrt{k}$ clusters each contain at least $n/k$ points, then with high probability each such large cluster is represented in the sample; estimating robustness against the original dataset then finds robust centers from these clusters. Second, if the remaining $k - \sqrt{k}$ clusters together contain at most $n/t$ points for some constant $t$ (e.g. $t = 4$), then the large clusters account for at least $(1 - 1/t)n$ points and, by the pigeonhole principle, at least one large cluster must contain $\Omega(n/\sqrt{k})$ points. Sampling $\widetilde{O}(\sqrt{k})$ points uniformly at random then ensures (with high probability) that we include a point from this large cluster, and we can verify its robustness against the full dataset.

Finally, another technical ingredient is a de-amortization technique that converts the *amortized* update time of a Monte Carlo algorithm [Motwani and Raghavan, 1999] into a *worst-case* update time. We use it to obtain a worst-case guarantee for the diameter problem. We also note that a similar de-amortization framework (although not directly applicable to our problems/setting) was recently proposed by Bernstein et al. [2021] for dynamic spanner and maximal matching problems.

## 1.2 Preliminaries

In this section, we introduce some notation, the problems, and necessary definitions that will be used throughout the paper.

Let $P \subseteq \mathbb{R}^d$ be a set of points in $d$ dimensions. Define $\mathrm{dist}(p, p')$ as the distance between two points $p, p' \in P$. Let $\mathrm{dist}(p, C) = \min_{c \in C} \mathrm{dist}(p, c)$ be the minimum distance between $p \in P$ and $C \subseteq P$, and $F(c, P) = \max_{p \in P} \mathrm{dist}(c, p)$ be the furthest neighbor from a point $c \in P$. With a slight abuse of notation, we may use $F(c, P)$ to refer to $\arg\max_{p \in P} \mathrm{dist}(c, p)$ as well. Further, denote $B(p, r)$ as the ball of radius $r$ centered at $p$, $\mathrm{MEB}(P)$ as the minimum enclosing ball of $P$, and $\mathrm{conv}(P)$ as the convex hull of $P$. Finally, let $n$ be the total length of the stream given in input. We remark that $n$ is used only for the analysis, our algorithms will not need to know $n$.

We are now ready to state the exact definition of the problems being studied.

**Definition 1.3** (Diameter Problem). *Given a set of points $P$, the goal of the algorithm is to return an estimate diameter that is as close as possible to $\mathrm{diam}(P) = \max_{p, p' \in P} \mathrm{dist}(p, p')$.*

**Definition 1.4** ($k$-Center Problem). *Given a set of points $P$, the goal of the algorithm is to choose a set of $k$ points $C \subseteq P$ such that $\max_{p \in P} \mathrm{dist}(p, C)$ is minimized.*

The $1$-center problem corresponds to the problem of minimum enclosing ball. For simplicity, we state the fully-dynamic version of the $k$-center problem below. The diameter problem is analogous with $OPT(P)$ being the diameter of $P$ instead.

**Definition 1.5** (Dynamic $\alpha$-Approximate $k$-Center against an Adaptive Adversary). *Given a set of points $P$ and a sequence of $n$ updates (insertions and deletions), the goal of the algorithm is to return at any point in time an answer $\tilde{d}$ such that $\tilde{d} \in [OPT(P), \alpha \cdot OPT(P)]$, for a constant $\alpha \geq 1$, with $OPT(P)$ being the radius of an optimal solution to the $k$-center problem.*

*The failure probability of the algorithm, i.e., the probability that at any point in time the algorithm returns a wrong answer, should be at most $\delta$, for arbitrary $\delta \in (0, 1)$. Moreover, the approximation factor and the update time should hold with a sequence of updates that are chosen by an adaptive adversary who sees the random bits used by the algorithm.*

## 2 Robust representative of a point set

As mentioned in the introduction, if we do not need adaptiveness, then we can obtain a 2-approximation for the diameter by maintaining the maximum distance from a fixed point in the dataset. The key intuition here is that each point in the dataset is (approximately) representative of the entire set for the purposes of calculating the diameter. Our first ingredient to develop our algorithms is a characterization of the property for a point $x \in \mathbb{R}^d$ of being *representative* of the set $P$, even if $x \notin P$.

**Definition 2.1** (Representative Point). *A point* $x \in \mathbb{R}^d$ *is* representative *of a point set* $P$ *if* $\max_{y \in P} \operatorname{dist}(x, y) \leq \operatorname{diam}(P)$.

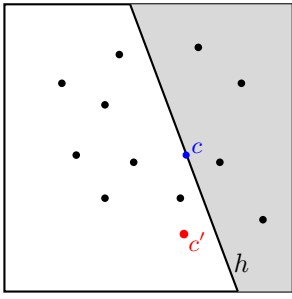

Figure 1: Representativeness and robustness in a point set of size 12. The red point, $c'$, is representative but not robust. The blue point, $c$, not part of the dataset, is both representative and $\epsilon$-robust for $\epsilon = 1/4$. In fact, one can verify that any subset of 9 or more points from the original dataset will still contain $c$ in its convex hull. This is because $c$ has Tukey depth 4, i.e., every halfspace passing through $c$ contains at least 4 points. The shaded region indicates one such halfspace. (See Section 2.1 for corresponding definitions.)

By definition, if we can maintain the maximum distance from a representative point, we always have an approximation for the diameter. In the dynamic setting however, the underlying set $P$ is going through updates and as such, any fixed point may not remain representative after some updates. Indeed, while any $x \in P$ satisfies the property of being representative, if the adversary removes $x$, we will need to choose a new point. To deal with this issue, we characterize the *robustness* of a representative point with respect to the point set.

**Definition 2.2** ($\epsilon$-Robust Representative Point). *A point* $x \in \mathbb{R}^d$ *is an* $\epsilon$-*robustly representative of* $P$ *if* $x$ *is a representative point of all* $P'$ *that satisfies* $|P' \cap P| > (1 - \epsilon)|P|$, *for an* $\epsilon \in (0, 1)$.

In the above definition, the same point $x$ should be representative of every set $P'$ that include at least a $(1 - \epsilon)$-fraction of points in $P$. Hence, $P'$ might also include an arbitrary number of points that are not in $P$. We refer to Fig. 1 for an illustration of these concepts.

The following lemma demonstrates the importance of an $\epsilon$-robust representative point. It essentially states that maintaining the maximum distance from a robust representative point gives a 2-approximation of the diameter, even when the original set undergoes any number of insertions and up to $\lfloor \epsilon |P| \rfloor$ deletions.

**Lemma 2.3.** *Assume that* $x \in \mathbb{R}^d$ *is an* $\epsilon$-*robustly representative of* $P$. *For any set* $P'$ *with* $|P' \cap P| > (1 - \epsilon)|P|$, *we have* $\operatorname{diam}(P')/2 \leq \max_{y \in P'} \operatorname{dist}(x, y) \leq \operatorname{diam}(P')$.

### 2.1 Robust reprentatives and centerpoints

While the definition of robust representatives is fairly natural given our problem, it is not clear whether such points exists and, importantly, whether they can be found efficiently. In this section, we show a construction of robustly representative points using so-called centerpoints, points sufficiently "deep" in the data. To formalize this, we need to introduce the notion of Tukey depth.

**Definition 2.4** (Tukey Depth [Tukey, 1975]). *Given a set of* $n$ *points* $P \in \mathbb{R}^{n \times d}$, *the Tukey's depth of a point* $x$ *is the smallest number of points in any closed halfspace that contains* $x$.

It is a known fact that every set of points (with no duplicates) always has a point, which need not be one of the data points, of Tukey depth at least $n/(d + 1)$, and this may be tight. Such a point can be thought of as a higher-dimensional median of the point set; in $d = 1$ the median has in fact Tukey depth at least $n/2$. It is useful to relate the Tukey depth of a point with the size of the point set, as captured by the following definition.

**Definition 2.5** ($\alpha$-centerpoint). *An* $\alpha$-*centerpoint* $c$ *of a point set of size* $n$ *is a point that has Tukey depth at least* $\lceil \alpha n \rceil$, *for any* $\alpha \in [\frac{1}{n}, \frac{1}{d+1}]$.

To gain familiarity with the definition, the one-dimensional median is a $1/2$-centerpoint. In higher dimensions, an $\alpha$-centerpoint point essentially divides a set of points in two subsets such that the smaller part has at least an $\alpha$ fraction of the points.

There are several known algorithms to compute a centerpoint with runtime depending on the quality of the centerpoint and the number of dimensions. Ideally, we would like to find a $(1/d+1)$-centerpoint, but this requires $O(n^{d-1} + n \log n)$ time by Chan [2004b], which is claimed to be probably optimal. The work of Clarkson et al. [1993] shows that an approximate $1/\text{poly}(d)$-centerpoint can be computed in $\text{poly}(d)$ time with high probability. Their algorithm is based on iteratively replacing sets of $d + 2$ points with their Radon point, which is any point that lies in the intersection of the convex hulls of these sets and is due to Radon [1921]. The fastest known algorithm is due to [Har-Peled and Jones, 2020] and has the following randomized guarantee.

**Theorem 2.6** (Theorem 3.9 of [Har-Peled and Jones, 2020]). *Given an arbitrary set $P$ of $n$ points in $\mathbb{R}^d$, there is a randomized Monte Carlo algorithm that computes a $\frac{1}{3d^2}$-centerpoint of $P$ in $O(d^7 \log^3 d \log^3 \varphi^{-1})$ time with probability at least $1 - \varphi$.*

We next show that an $\alpha$-centerpoint satisfies two key requirements: (i) it is representative, i.e., it leads to a 2 approximation of the diameter, and (ii) it is robust, i.e., it continues to provide a 2-approximation even after $\Omega(\alpha|P|)$ deletions. We begin by proving the following lemma which shows that any point in the convex hull is representative.

**Lemma 2.7.** *Let $c$ be a point in the convex hull of $P$. Then, $F(c, P) \in [\frac{\text{diam}(P)}{2}, \text{diam}(P)]$.*

Let $r(\cdot)$ denote the radius of a set. We immediately obtain the following corollary.

**Corollary 2.8.** *Let $c$ be a point in the convex hull of $P$. Then, $F(c, P) \in [r(MEB(P)), 2r(MEB(P))]$.*

**Lemma 2.9.** *Any point $c$ with Tukey depth at least $1$ with respect to a set $P$ lies in its convex hull.*

Combining Lemma 2.7 and Lemma 2.9 proves that any point with Tukey depth $1$ is representative. Next, we show that the Tukey depth of a centerpoint for $P$ can be lower bounded, even if $P$ goes through $\varepsilon|P|$ adversarial updates.

**Lemma 2.10.** *Let $P$ be a set of $n$ points and $c$ be an $\alpha$-centerpoint of $P$. For any set $P'$, the point $c$ has Tukey depth at least $\lceil \alpha n - t \rceil$ with respect to $P'$ where $t = |P \setminus P'|$.*

*In the special case where $|P' \cap P| > (1 - \varepsilon)|P|$ is satisfied, for $\varepsilon < \alpha$, the point $c$ has Tukey depth at least $1$ with respect to $P'$.*

Combining Lemma 2.7, Lemma 2.9, and Lemma 2.10, we obtain the following.

**Corollary 2.11.** *Any $\alpha$-centerpoint of $P$ is $\epsilon$-robustly representative for any $\epsilon < \alpha$.*

## 3  Dynamic algorithm for the diameter problem

We are now ready to explain how to compute and maintain a 2-approximation to the diameter of a dynamic point set $P$, against an adaptive adversary. Let $\varepsilon = 1/4d^2$ be a parameter, $\delta$ be an upper bound on the desired failure probability, and $t$ denote the current timestep, i.e., the number of updates occurred so far.

If the size of the current point set is small, say at most $d^4$, we can obtain a 2-approximation without a centerpoint. The algorithm picks an arbitrary point in $P$ and computes its furthest neighbor in $O(d^5)$

When the size of $P$ is larger than $d^4$, we compute an $\alpha$-centerpoint $c$ that will be kept as a representative point until roughly $\alpha|P|$ deletions occur. Once we have a centerpoint $c$, we maintain $F(c, P)$ using a balanced BST (e.g., implemented as an AVL tree) that contains the distances of all points from the centerpoint. We then use a counter to keep track of the number of deletions that $P$ has undergone since the computation of $c$, so that another centerpoint will be computed as soon as $c$ stops being representative. We give a more detailed pseudocode below: queries are handled by Algorithm 1, insertions by Algorithm 2, and deletions by Algorithm 3.

---

**Algorithm 1** APPROXIMATEDIAMETERQUERY($P, d, \varepsilon, t, \delta$)

---
1: **if** $P$.counter $> 0$ **then**        ▷ The previously computed centerpoint is still representative
2:     **return** $F(P\text{.centerpoint}, P)$

3: **if** $|P| \leq d^4$ **then**                                      ▷ Use an arbitrary point
4:     Pick an arbitrary point $p \in P$
5:     **return** $F(p, P)$

6: $P$.counter $\leftarrow \varepsilon|P|$                              ▷ Compute a new centerpoint
7: $P$.centerpoint $\leftarrow$ Find $1/3d^2$-centerpoint of $P$ via Theorem 2.6 with $\varphi = \delta/2t^2$
8: $P$.distances $\leftarrow$ build a balanced BST on $\bigcup_{p \in P} \text{dist}(c, p)$
9: **return** $F(P\text{.centerpoint}, P)$

---



---

**Algorithm 2** APPROXIMATEDIAMETERINSERTION($P, p$)

---
1: insert $p$ into $P$
2: **if** $P$.counter $> 0$ **then**
3:     insert $\text{dist}(P\text{.centerpoint}, p)$ into $P$.distances

---



---

**Algorithm 3** APPROXIMATEDIAMETERDELETION($P, p$)

---
1: delete $p$ from $P$
2: **if** $P$.counter $> 0$ **then**
3:     delete $\text{dist}(P\text{.centerpoint}, p)$ from $P$.distances
4:     $P$.counter $\leftarrow P$.counter - 1

---

We establish the approximation guarantee achieved by our algorithm in the next lemma.

**Lemma 3.1.** *Given a sequence $S$ of $n$ updates by an adaptive adversary, with each update being either an insertion or a deletion of a point, let $P_t$ be the set obtained after the first $t$ updates. The algorithm* APPROXIMATEDIAMETERQUERY *(Algorithm 1) returns a 2-approximate diameter of $P_t$, at any point in time $t \in [n]$, with probability at least $1 - \delta$.*

Next, we prove an amortized bound on the cost of each operation performed by our algorithm.

**Lemma 3.2.** *Given a sequence $S$ of $n$ possibly adversarial updates, let $P_t$ be the set obtained after applying the first $t$ updates. The amortized update time per operation is $O(d^5 \log^3 d \log^3(n/\delta))$.*

We are now ready to prove a worst-case bound on the update time. By the discussion above, it is enough to guarantee that at all times a centerpoint is available, so that we do not need to compute one from scratch before answering a query. We explain how to maintain a valid centerpoint at all times when $|P_t| = \Omega(d^4 \log^{1.5} d \log^{1.5} \varphi^{-1})$. If the latter assumption is not respected, we can answer each query on the fly in $O(d^5 \log^{1.5} d \log^{1.5} \varphi^{-1})$. To build some intuition, consider a centerpoint $c$ computed at time $t$. Point $c$ will be valid for the next $\varepsilon|P_t|$ updates. Rather than waiting $\varepsilon|P_t|$ timesteps, we initiate the computation of the next centerpoint $c'$ at time $t + 3/4 \cdot \varepsilon|P_t|$. The computation of $c'$ is being spread over the next $1/4 \cdot \varepsilon|P_t|$ iterations in which $c$ is still valid. This process can be implemented as shown in Algorithm 4, where each iteration of the for loop corresponds to an update.

---

**Algorithm 4** DE-AMORTIZEDCENTERPOINTCOMPUTATION($d, \varepsilon$)

---

1: centerpoint ← ∅
2: expiration_time ← 0           ▷ When the current centerpoint stops being representative
3: update_time ← 0                 ▷ When to start computing the next centerpoint

4: next_centerpoint ← ∅
5: next_expiration_time ← 0

6: **for** $t = 1, \ldots n$ **do**
7:     **if** $t$ equals expiration_time **then**              ▷ The next centerpoint takes over
8:        centerpoint ← next_centerpoint
9:        expiration_time ← next_expiration_time
10:    **if** $t$ equals update_time **then**       ▷ The computation of the next centerpoint starts
11:        **do** in the background within $T$ timesteps (cf. Lemma 3.3)
12:          Create a copy of $P_t$ (filter newer points)
13:          next_centerpoint ← $(1/3d^2)$-centerpoint of $P_t$ with $\varphi = \delta/2t^2$
14:          Compute all distances from next_centerpoint (include newer points)
15:        next_expiration_time ← $t + \varepsilon|P_t|$
16:        update_time ← $t + {}^3\!/4 \cdot \varepsilon|P_t|$

---

**Lemma 3.3.** *The worst-case update time of Algorithm 1 combined with the centerpoint computation of Algorithm 4 is $O(d^5 \log^{1.5} d \log^{1.5}(n/\delta))$. This update time holds deterministically, regardless of the adversarial nature of the updates.*

Finally, Theorem 1.1 follows from combining Lemma 3.1 with Lemma 3.3.

## 4   Dynamic algorithm for $k$-center clustering

In this section, we describe our dynamic algorithm for the $k$-center problem. Our approach is to identify as a measure of robustness the number of points within a certain radius from each candidate center. Since this would be computationally expensive to calculate exactly, we use random sampling to either find candidate centers or estimate the density of each candidate. We will prove that the point with the highest density provides a robust center. We then cluster all points close to such a center (breaking ties assigning always to the lexicographically smaller). The dynamic maintenance of each cluster will follow from the robustness of the chosen center, defined as follows.

**Definition 4.1** (($\ell, \beta$)-robust center). *Given a point set $P \subset \mathbb{R}^{n \times d}$, a point $x \in \mathbb{R}^d$ is an ($\ell, \beta$)-robust center of $P$ if $|P \cap B(x, 2(1+\varepsilon)^\ell)| \geq \beta$, for $\ell \in [L]$, $\beta \in [n]$, and some error parameter $\varepsilon > 0$.*

Our approach will reduce the dynamic $k$-center problem to maintaining an ordered sequence of at most $k$ instances $\mathcal{I}_1, \ldots, \mathcal{I}_k$, where: (1) each instance is associated with a robust-center, (2) instance $\mathcal{I}_i$ is constructed recursively after completing instances $\mathcal{I}_1, \ldots, \mathcal{I}_{i-1}$, and (3) if, at any point during the dynamic process, an instance $\mathcal{I}_i$ must be reconstructed, then all subsequent instances $\mathcal{I}_i, \ldots, \mathcal{I}_k$ are rebuilt as well. We first introduce a static algorithm and then discuss its robust dynamization. The pseudocode of all our algorithms is included in the appendix.

**Static $k$-center algorithm**    Let $\rho = \frac{\max_{p,q \in P} \text{dist}(p,q)}{\min_{p,q \in P} \text{dist}(p,q)}$ denote the *spread ratio* of the point set $P \subset \mathbb{R}^d$. We consider guesses $\ell \in [L]$, where $L = \lceil \log_{(1+\varepsilon)} \rho \rceil$, for the optimal radius $OPT(P)$, and run the following procedure in parallel for each guess.

For each guess $(1+\varepsilon)^\ell$, we uniformly sample a set $S^\ell$ of $O(k \log L \log(n/\delta))$ points from $P$; the actual size of the $S^\ell$ may be optimized based on the number of unclustered points. For each sample point $s \in S^\ell$, we compute how many other sample points lie within distance $2(1+\varepsilon)^{\ell+1}$, i.e., $\left|B(s, 2(1+\varepsilon)^{\ell+1}) \cap S^\ell\right|$. The sample point $s^\ell$ with the highest count is chosen as a center. If $\ell$ is a good guess (i.e., close to $OPT(P)$), then there is at least one optimal large cluster that contains at least $n/k$ points, which will provide a $(\ell, \Omega(n/k))$-robust center.

Once the sample point $s^\ell$ with the largest neighborhood is selected, we gather all points in $P$ within $B(s_i^\ell, 4(1+\varepsilon)^{\ell+1})$, where the radius is doubled to ensure complete coverage. We then define the

corresponding cluster $\mathfrak{C}_i^\ell$ with $c_i^\ell := s^\ell$ as $P \cap B(c^\ell, 4(1+\varepsilon)^{\ell+1})$, and remove its points from $P$ before recursing. If $P$ becomes empty within $k$ iterations, we have a *good* guess; otherwise, the guess is marked as *low-value*. We always return the smallest good guess. See the appendix for the pseudocode.

**Robust dynamization**   We now adapt the static algorithm to the dynamic setting. We maintain $L$ parallel copies of the dataset—one per guess—denoted $P^\ell$. For each cluster $\mathfrak{C}_i^\ell$, we maintain two counters: $n_i^\ell$ tracks the cluster size, and $m_i^\ell$ counts how many updates can be performed before rebuilding the cluster. When $m_i^\ell$ hits zero, we rebuild $\mathfrak{C}_i^\ell$ and all subsequent clusters for guess $\ell$.

On insertion of a point $p$, we add it to each $P^\ell$. If $p$ falls within a current cluster $\mathfrak{C}_i^\ell$, we add it to such a cluster and increment $m_i^\ell$ if $p$ is close to its center. If $p$ is not covered by any existing cluster, we add it to the remainder set $Q^\ell$. These remainder sets allow us to detect increases in $OPT(P)$ over time. In such cases, we extract the solution from a clustering associated with a higher guess.

Deletions are handled similarly. When a point $p$ is deleted, we remove it from each $P^\ell$ and determine whether it belonged to a cluster $\mathfrak{C}_i^\ell$. If it has distance at most $2(1+\varepsilon)^{\ell+1}$ from $c_i^\ell$, we decrement $m_i^\ell$ and rebuild if needed. If $p$ was part of the remainder set $Q^\ell$, we remove it from $Q^\ell$ and check whether to resume a previously suspended clustering for guess $\ell$. This is because the deletion may reduce $OPT(P)$, making a previously invalid guess $(1+\varepsilon)^\ell$ now feasible. If so, we resume clustering from the iteration it was halted.

**Handling small guesses**   A subtle but important issue arises when the guess $(1+\varepsilon)^\ell$ is significantly smaller than $OPT(P)$. In such cases, all sample neighborhoods $B(s, 2(1+\varepsilon)^{\ell+1})$ may contain fewer than $O(\varepsilon^{-2} \log n)$ sample points, making them too sparse to be useful. To address this, whenever we observe that all sample neighborhoods in $S^\ell$ are too sparse, we suspend the clustering process. We resume it only when a sample point in $S^\ell$ gains enough neighbors (at least $O(\varepsilon^{-2} \log n)$), indicating that the guess may now be viable.

Let $C^* = \{c_1^*, \ldots, c_k^*\}$ be the set of optimal centers and let $\mathfrak{C}^* = \{\mathfrak{C}_1^*, \ldots, \mathfrak{C}_k^*\}$ be the set of optimal clusters. The approximation achieved by our algorithm is established via the following invariants.

**Definition 4.2.** *At all times, our algorithm satisfies the following two invariants:*

- **disjointedness**: *For any $i \in [k-1]$, the set $B(c_i, 4(1+\varepsilon)OPT(P)) \cap C_j = \emptyset, j > i$.*

- **closeness**: *for every $i \leq k$, it holds that $\mathrm{dist}(c_i^\ell, \mathfrak{C}_i^*) \leq 2(1+\varepsilon)OPT(P)$, that is, there exists a point $p$ from $\mathfrak{C}_i^*$ that is "close" to our center.*

The bound on the update time follows the same line of reasoning as that developed for the diameter. We refer to the appendix for the details of our algorithm for $k$-center.

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

# A Further related work

Concerning the diameter problem in high-dimensions, a better-than-two approximation can be achieved in the oblivious adversary setting. A long line of work [Borodin et al., 1999, Egecioglu and Kalantari, 1989, Finocchiaro and Pellegrini, 2002, Goel et al., 2001, Indyk, 2000, 2003] culminated in a fully dynamic $(1 + \epsilon)$-approximate algorithm with amortized update time $\tilde{O}(n^{\frac{1}{(1+\epsilon)^2}})$ by Indyk [2003]. A $(\sqrt{2} + \epsilon)$-approximation with $O(d \log n)$ amortized update time is achieved by the works of [Agarwal and Sharathkumar, 2015, Chan and Pathak, 2014].

While our focus is on high-dimensional spaces, the diameter problem is well-understood for low, fixed dimensions. Indeed, it is relatively simple to obtain a $(1 + \epsilon)$-approximation by maintaining $1/\epsilon^{O(d)}$ different directions [Agarwal et al., 2004, Agarwal and Yu, 2007, Chan, 2004a, Zarrabi-Zadeh, 2011]. The number of dimensions can often be reduced by using dimensionality reduction, however, the above approaches struggle even with $\log n$ dimensions, due to the exponential dependency on $d$, which renders them inapplicable in practical scenarios.

Another line of work considered the closely related problem of minimum enclosing ball (MEB). It corresponds to the 1-center problem, and a $c$-approximate MEB directly implies a $(\sqrt{2} \cdot c)$-approximate diameter, for $d$ large enough. A number of streaming and dynamic MEB algorithm have been developed [Egecioglu and Kalantari, 1989, Agarwal and Sharathkumar, 2015, Chan and Pathak, 2014, Zarrabi-Zadeh and Chan, 2006]. The best known dynamic MEB algorithm is a 1.22-approximation by Chan and Pathak [2014] that builds upon the streaming algorithm of Agarwal and Sharathkumar [2015], which is based on coresets [Bădoiu and Clarkson, 2008, Kumar et al., 2003].

The k-center problem was first studied by Charikar et al. [1997] in the incremental setting (where the points are inserted and the goal is to maintain an approximate solution), who introduced the doubling algorithm, achieving an 8-approximation with an amortized update time of $O(k)$. Subsequent work by Chan et al. [2018] presented a randomized algorithm that maintains a $(2 + \varepsilon)$-approximation, with $O(k^2/\epsilon \log \rho)$ amortized time per update, with $\rho$ being the aspect ratio. For low-dimensional spaces, the work of [Goranci et al., 2021] provides a $(2 + \varepsilon)$-approximate deterministic algorithm with update time $(2/\varepsilon)^{O(d+\lg \lg \rho)}$. Most recently, Bateni et al. [2023] extended these results to general metric spaces, providing an optimal $(2 + \varepsilon)$-approximate randomized algorithm with amortized update time $O(k \cdot \text{poly} \log(n, \Delta))$. Additionally, they developed a deterministic algorithm with approximation factor $O(\min\{k, \frac{\log(n/k)}{\log \log n}\})$. Another closely related line of work on *consistent k-center clustering* has focused on algorithms for $k$-center with small recourse [Łącki et al., 2024, Bhattacharya et al., 2024a,b, Forster and Skarlatos, 2025].

There is a large body of work on dynamic problems against an adaptive adversary. These include convex hull in two and three dimensions [Chan, 2010], geometric set cover [Chan and He, 2021], and numerous graph problems such as matching [Wajc, 2020, Roghani et al., 2022], spanning forest [Nanongkai and Saranurak, 2017, Nanongkai et al., 2017], spanners [Bernstein et al., 2020], incremental single-source shortest path [Probst Gutenberg et al., 2020, Alokhina and van den Brand, 2024], decremental single source shortest path [Bernstein and Chechik, 2016, Chuzhoy and Khanna, 2019, Gutenberg and Wulff-Nilsen, 2020], general single source shortest-path [Karczmarz et al., 2022], and all pairs shortest paths [Evald et al., 2021, Beimel et al., 2022, Chuzhoy and Zhang, 2023].

Several notions of black-box and white-box adaptive adversaries have been studied in the streaming model [Ben-Eliezer et al., 2022, Kaplan et al., 2021, Cohen et al., 2022]. The white-box streaming model was recently introduced by Ajtai et al. [2022]. Research in the white-box model has focused on graph matching and heavy hitters (See e.g., [Ajtai et al., 2022, Alokhina and van den Brand, 2024, Stoeckl, 2023, Feng and Woodruff, 2023, Feng et al., 2024]).

There are also a number of other algorithms for distance and clustering related problems in various models of computations, including dynamic [Chan et al., 2018, Schmidt and Sohler, 2019, Goranci et al., 2021, Chan et al., 2024, Biabani et al., 2024], streaming Ceccarello et al. [2019], De Berg et al. [2023], sliding window [Cohen-Addad et al., 2016, Woodruff and Zhou, 2022, Pellizzoni et al., 2022], and massively parallel computation Malkomes et al. [2015], Bateni et al. [2021], Coy et al. [2023], Czumaj et al. [2025]. These results however do not apply to our more stringent setting of high dimensions against an adaptive adversary.

# B   Missing Proofs and Pseudocode

## B.1   Robust representative

**Lemma 2.3.** *Assume that $x \in \mathbb{R}^d$ is an $\epsilon$-robustly representative of $P$. For any set $P'$ with $|P' \cap P| > (1 - \epsilon)|P|$, we have $\mathrm{diam}(P')/2 \leq \max_{y \in P'} \mathrm{dist}(x, y) \leq \mathrm{diam}(P')$.*

*Proof.* Recall $F(x, P) := \max_{y \in P} \mathrm{dist}(x, y)$ from Section 1.2. The first inequality follows from the triangle inequality. For a diametral pair $y, z \in P'$ with $\mathrm{dist}(y, z) = \mathrm{diam}(P')$, we have

$$\mathrm{diam}(P') = \mathrm{dist}(y, z) \leq \mathrm{dist}(y, x) + \mathrm{dist}(z, x) \leq 2 \max_{y \in P'} \mathrm{dist}(x, y) = 2F(x, P).$$

Then, the second inequality follows from the fact that $x$ is a robust representative point of $P$ and $|P' \cap P| > (1 - \epsilon)|P|$. That is, $x$ is a representative point of $P'$, so $F(x, P') \leq \mathrm{diam}(P')$ holds. $\square$

**Lemma 2.7.** *Let $c$ be a point in the convex hull of $P$. Then, $F(c, P) \in [\frac{\mathrm{diam}(P)}{2}, \mathrm{diam}(P)]$.*

*Proof.* If $c \in P$, then the claim follows trivially. Suppose that $c \notin P$ and let $p \in P$ be the furthest point from $c$ such that $p = F(c, P)$. The inequality $F(c, P) \geq \mathrm{diam}(P)/2$ follows from the triangle inequality. We now prove that $F(c, P) \leq \mathrm{diam}(P)$ by showing that there is a point $c'$ such that: (i) $c' \in P$ and (ii) $\mathrm{dist}(c', p) \geq \mathrm{dist}(c, p)$. This will finish the proof since $\mathrm{diam}(P) \geq \mathrm{dist}(c', p)$ by definition.

Observe that the $\ell_2$-distance function $\mathrm{dist}(x, p)$ is continuous and convex with respect to $x \in \mathrm{conv}(P)$ for a fixed point $p$. The convex hull is non-empty because it contains $c$ and is compact because it is a closed and bounded set. Moreover, by the extreme value theorem, we know that the function $\mathrm{dist}(x, p)$ attains its maximum on any compact set. Observe however that $\mathrm{dist}(x, p)$ is a convex function, which means it attains its maximum value over the convex hull on one of the vertices. Since $c \in \mathrm{conv}(P)$, this means that there exists at least one vertex $c'$ such that $\mathrm{dist}(c', p) \geq \mathrm{dist}(c, p)$. Since any vertex of $\mathrm{conv}(P)$ is in $P$, this proves that

$$F(c, P) = \mathrm{dist}(c, p) \leq \mathrm{dist}(c', p) \leq F(c', P) \leq \mathrm{diam}(P). \qquad \square$$

**Corollary 2.8.** *Let $c$ be a point in the convex hull of $P$. Then, $F(c, P) \in [r(\mathit{MEB}(P)), 2r(\mathit{MEB}(P))]$.*

*Proof.* The ball $B$ centered at $c$ with radius $F(c, P)$ encloses all the points. Moreover, by the above proof, a ball that encloses both $c'$ and $F(c, P)$ must have radius at least $F(c, P)/2$. $\square$

**Lemma 2.9.** *Any point $c$ with Tukey depth at least $1$ with respect to a set $P$ lies in its convex hull.*

*Proof.* Suppose that $c$ lies outside of the convex hull of $P$, i.e., $c$ and $\mathrm{conv}(P)$ are disjoint. Since $\mathrm{conv}(P)$ is a closed and compact convex set and so is $\{c\}$, we can apply the following version of the hyperplane separation theorem: If two disjoint convex sets are closed and at least one of them is compact, then there is a hyperplane $h$ that strictly separates them. Then, we observe that the existence of $h$ contradicts the fact that $c$ has Tukey depth at least $1$. This is because the halfspace defined by $h$ from $c$'s side contains $c$ and has no point from $P$, which would mean that $c$ has Tukey depth zero. $\square$

**Lemma 2.10.** *Let $P$ be a set of $n$ points and $c$ be an $\alpha$-centerpoint of $P$. For any set $P'$, the point $c$ has Tukey depth at least $\lceil \alpha n - t \rceil$ with repsect to $P'$ where $t = |P \setminus P'|$.*

*In the special case where $|P' \cap P| > (1 - \varepsilon)|P|$ is satisfied, for $\varepsilon < \alpha$, the point $c$ has Tukey depth at least $1$ with respect to $P'$.*

*Proof.* Since $c$ is an $\alpha$-centerpoint of $P$, any closed halfspace $h$ containing $c$ has at least $\lceil \alpha n \rceil$ points from $P$. Then, $h$ contains at least $\lceil \alpha n \rceil - t$ points from $P'$, since at most $t$ points from $P$ are not in $P'$. Therefore, $c$ has Tukey depth at least $\lceil \alpha n \rceil - t = \lceil \alpha n - t \rceil$ with respect to $P'$. For the second part of the Lemma, observe that if $\varepsilon < \alpha$ then

$$\lceil \alpha n - t \rceil \geq \alpha n - |P \setminus P'| = \alpha n - (n - |P \cap P'|) = \alpha n - n + (1 - \varepsilon)n = \alpha n - \varepsilon n > 0.$$

Since $\lceil \alpha n - t \rceil$ is an integer, it follows that it is at least $1$. $\square$

## B.2 Diameter

**Lemma 3.1.** *Given a sequence $S$ of $n$ updates by an adaptive adversary, with each update being either an insertion or a deletion of a point, let $P_t$ be the set obtained after the first $t$ updates. The algorithm* APPROXIMATEDIAMETERQUERY *(Algorithm 1) returns a 2-approximate diameter of $P_t$, at any point in time $t \in [n]$, with probability at least $1 - \delta$.*

*Proof.* Fix an arbitrary $t \in [n]$. We prove that our algorithm returns a 2-approximate diameter by either computing it statically (Line 5) or by using a centerpoint (Lines 2 and 9). In the first case, when the algorithm returns $F(p, P_t)$ for some $p \in P_t$, a 2-approximation of the current diameter immediately follows.

If a centerpoint $c$ is available, then the algorithm returns $F(c, P_t)$. Suppose that $c$ was computed at time $t_0$ for $P_{t_0}$. By Theorem 2.6, $c$ is an $(1/3d^2)$-centerpoint of $P_{t_0}$, with probability at least $1 - \varphi$. By Corollary 2.11, $c$ is also an $\varepsilon$-robust representative of $P_{t_0}$ where $\varepsilon = 1/4d^2$. We now show that $c$ is a representative point of the current point set. To this end, we prove that $|P_t \cap P_{t_0}| > (1 - \varepsilon)|P_{t_0}|$. It is easy to verify that the following invariant holds

$$|P_t \cap P_{t_0}| = |P_{t_0}| - |P_t \setminus P_{t_0}| \geq |P_{t_0}| - (\varepsilon|P_{t_0}| - P.counter) > (1 - \varepsilon)|P_{t_0}|,$$

since, any time that a point is deleted, $P.counter$ is decremented by one and $P.counter$ can only take on values in $\{1, \ldots, \varepsilon|P_{t_0}|\}$. With the above, $P_t$ satisfies Definition 2.2 and thus $c$ is a representative point of $P_t$. We conclude that by Lemma 2.3, $F(c, P_t)$ gives a two approximation to $\mathrm{diam}(P_t)$.

It remains to bound the probability of failure. The only randomness used is in Theorem 2.6, and if the algorithm succeeds, the 2-approximation holds for all updates, even adaptive ones. Specifically, the success of a query at time $t$ depends only on whether $c$ was a $(1/3d^2)$-centerpoint of $P_{t_0}$. Since Theorem 2.6 uses fresh randomness independent of $P_{t_0}$, the adversary cannot influence its outcome. In particular, the adversary cannot retroactively change $P_{t_0}$. This means that each centerpoint computation fails with probability at most $\varphi$, independently of the adversarial nature of the updates. By taking a union bound over the at most $n$ centerpoint computations, we have

$$\Pr[\text{Algorithm 1 fails}] \leq \Pr[\exists \text{ ``bad'' centerpoint}] \leq \sum_{t=1}^{n} \frac{\delta}{2t^2} \leq \delta.$$

$\square$

**Lemma 3.2.** *Given a sequence $S$ of $n$ possibly adversarial updates, let $P_t$ be the set obtained after applying the first $t$ updates. The amortized update time per operation is $O(d^5 \log^3 d \log^3(n/\delta))$.*

*Proof.* Note that insertions and deletions are handled in $O(d \log n)$ time by Algorithm 2 and Algorithm 3. When the adversary issues a query at time $t \in [n]$, there are three possible cases. If $P.counter$ is not zero, then the query time is given by a simple $O(1)$-time lookup operation (Line 2). If $P.counter$ is zero and $|P_t| = O(d^4)$, then the query time is $O(d|P_t|) = O(d^5)$.

The most interesting case is when $P.counter$ is zero and $|P_t| > d^4$, which will be proved by induction on the number of centerpoints. For the base case, the first centerpoint computation takes $O(d^7 \log^3 d \log^3(n/\delta) + d|P_t| \log |P_t|)$ and can be amortized over the first $|P_t| \geq d^4$ insertions. For the inductive step, we amortize the computation of the $(i + 1)$-th centerpoint at time $t_{i+1}$ over the updates that occurred between time $t_{i+1}$ and time $t_i$, when the computation of the $i$-th centerpoint occurred. By induction, the computation of the $i$-th centerpoint is amortized over updates prior to $t_i$, so it does not affect the amortization on subsequent updates. The total computation to amortize is $O(d^7 \log^3 d \log^3(n/\delta) + d|P_{t_{i+1}}| \log |P_{t_{i+1}}|)$. Now, recall that $t_{i+1} - t_i \geq \varepsilon|P_{t_i}| \geq d^2$ by the centerpoint property, and observe that

$$\frac{|P_{t_{i+1}}|}{t_{i+1} - t_i} \leq \frac{|P_{t_i}| + t_{i+1} - t_i}{t_{i+1} - t_i} \leq O(1/\varepsilon),$$

proving the claimed amortized time. $\square$

**Lemma 3.3.** *The worst-case update time of Algorithm 1 combined with the centerpoint computation of Algorithm 4 is $O(d^5 \log^{1.5} d \log^{1.5}(n/\delta))$. This update time holds deterministically, regardless of the adversarial nature of the updates.*

*Proof.* The computation of a new centerpoint $c'$ at time $t$ happens sequentially in three steps:

1. Create of a copy of $P_t$ in $T_1$ timesteps, incurring $(|P_t| + T_1) \cdot O(d \log |P_t|)$ operations. To avoid conflicts, we add to each point an entry specifying the time in which it was inserted and another entry for the time in which it was deleted (if any). Moreover, we also maintain the points that were deleted in subsequent timesteps $t' \in [t, t + T_1]$ in a separate array, so that creating a copy of $P_t$ does not conflict with $P'_t$. The number of extra operations required is $O(T_1 d \log n)$, which we account for in the total complexity of this step.

2. Compute a new centerpoint $c'$ in $T_2$ timesteps, incurring $O(d^7 \log^3 d \log^3(n/\delta))$ operations.

3. Scan all of the points present to build an AVL tree with all distances from $c'$ in $T_3$ timesteps, incurring $(|P_t| + T_1 + T_2 + T_3) \cdot O(d \log |P_t|)$ operations. For this step, we use the same strategy as earlier to avoid creating any conflicts. The overhead incurred to handle extra copies of recently deleted and newly added points is at most $O(T_1 + T_2 + T_3 d \log n)$.

This computation starts at time $t$ and finishes at time $t + T = t + T_1 + T_2 + T_3$, and, at each timestep, up to $C \cdot d^5 \log^{1.5} d \log^{1.5}(n/\delta)$ operations can be performed, for some constant $C > 0$. For our worst-case update time guarantee to hold, we need to prove: (1) the computation of $c'$ finishes before the expiration of the current centerpoint; (2) no two centerpoint computations overlap. We prove both conditions by induction of the number of centerpoints. We assume that the computations of centerpoints are consecutive, since whenever the size of the dataset drops within $O(d^4)$, we can compute the diameter on the fly. For the base case, we observe that the first centerpoint is being computed when $|P_t| = \Theta(d^4)$, by the previous reasoning. Therefore, the algorithm can spend $O(d^4)$ iterations to compute it, while resorting to the on-the-fly computation before the centerpoint is ready.

For the inductive step, let $t_0$ be the point in time when the current centerpoint $c$ was computed. The total computation is $O(d^7 \log^3 d \log^3(n/\delta) + d|P_t| \log |P_t| + dT \log |P_t|)$. For (1), the computation of $c'$, which starts at $t = {}^{3\varepsilon}/4|P_{t_0}|$, should finish within $T \leq \varepsilon/4|P_{t_0}|$ iterations. This is verified since

$$\frac{d^7 \log^3 d \log^3(n/\delta) + d|P_t| \log |P_t| + dT \log |P_t|}{C \cdot d^5 \log^{1.5} d \log^{1.5}(n/\delta)} \leq \frac{d^2}{C} \log^{1.5} d \log^{1.5}(n/\delta) + \frac{|P_t| + T}{Cd^4} < \frac{\varepsilon|P_{t_0}|}{12},$$

where the last inequality follows from $|P_{t_0}| \geq \Omega(d^4 \log^{1.5} d \log^{1.5}(n/\delta))$ and $|P_t| + T < 3|P_{t_0}|$. For (2), we have that the computation of the successor of $c'$ will start at time $t + {}^{3\varepsilon}/4|P_t|$, which is always greater than $t + \varepsilon/4|P_{t_0}|$ because $|P_t| \geq (1 - \varepsilon)|P_{t_0}| > \frac{|P_{t_0}|}{2}$. $\qquad\square$

## B.3 $k$-center clustering

---
**Algorithm 5** INIT$(P, d, k, \varepsilon)$

---
**Input:** The input set $P$, the dimension $d$ and the error parameter $\epsilon$.

1: **for** $\ell = 0$ to $L$ **do**
2:      $P^\ell \leftarrow P$ is a copy of the point set $P$
3:      Initialize two sets $C^\ell \leftarrow \emptyset$ and $\mathfrak{C}^\ell \leftarrow \emptyset$ as the set of centers and clusters
4:      $C^\ell, Q^\ell, \mathfrak{C}^\ell \leftarrow$ CLUSTERING$(P_\ell, d, \varepsilon, \ell, 1)$
5:      The remainder set $Q^\ell$ is maintained as a *too low* certificate for the guess $(1 + \varepsilon)^\ell$
6: Let $\mathfrak{l} \in [L]$ be the smallest index for which $Q^{\mathfrak{l}}$ is empty
7: Return $C^{\mathfrak{l}}$ and $\mathfrak{C}^{\mathfrak{l}}$

---

### B.3.1 Proofs

### B.4 Correctness Analysis

This section is devoted to proving that our algorithm computes and maintains a $4(1+\varepsilon)$ approximation. We first introduce some notation. Recall that we consider $L$ guesses $(1+\varepsilon)^0, (1+\varepsilon)^1, \cdots, (1+\varepsilon)^L$ for the optimal $k$-center radius $OPT(P)$. Throughout, we fix the guess $(1 + \varepsilon)^{\ell^*} \leq OPT(P) < (1 + \varepsilon)^{\ell^*+1}$.

---

**Algorithm 6** CLUSTERING($X, d, \varepsilon, \ell, i \leftarrow 1$)

---

**Input:** The input set $X$, the dimension $d$, the error parameter $\epsilon$, the guess $\ell$.
**Output:** It (re-)constructs clusters $\mathfrak{C}_i^\ell, \cdots, \mathfrak{C}_k^\ell$, while clusters $\mathfrak{C}_1^\ell, \cdots, \mathfrak{C}_{i-1}^\ell$ are fixed.

  1: Let $Y \leftarrow X$ be a copy of $X$
  2: **while** $Y$ is not empty and $i \leq k$ **do**
  3:      Let $S_i^\ell$ be a set sampled u.a.r. from $Y$ (cf. Lemma B.3)
  4:      $c_i^\ell \leftarrow \arg\max_{q \in S_i^\ell} |B(q, 2(1+\varepsilon)^{\ell+1}) \cap (|Y| \text{ or } |S_i^\ell|)|$
  5:      **if** $|B(c_i^\ell, 2(1+\varepsilon)^{\ell+1}) \cap (|Y| \text{ or } |S_i^\ell|)| < (1-\varepsilon) \cdot \frac{|Y|}{4(k-i)}$ **then**
  6:          The guess $(1+\varepsilon)^\ell$ is small for $OPT(P)$
  7:          Break the while-loop
  8:      $\mathfrak{C}_i^\ell \leftarrow B(c_i^\ell, 4(1+\varepsilon)^{\ell+1}) \cap Y$, cluster around $c_i^\ell$
  9:      $m_i^\ell \leftarrow |B(q, 2(1+\varepsilon)^{\ell+1}) \cap Y|$, robustness of $c_i^\ell$
10:      $n_i^\ell \leftarrow |(B(c_i^\ell, 4(1+\varepsilon)^{\ell+1}) \cap Y)|$, number of points in $\mathfrak{C}_i^\ell$
11:      Add $c_i^\ell$ to $C^\ell$, and let $Y \leftarrow Y \backslash (B(c_i^\ell, 4(1+\varepsilon)^{\ell+1}) \cap Y)$
12:      $i \leftarrow i+1$
13: Return $C^\ell$, $Y$, and the clusters $\mathfrak{C}^\ell \leftarrow \{\mathfrak{C}_1^\ell, \cdots, \mathfrak{C}_i^\ell\}$

---

**Algorithm 7** CLUSTERINGINSERTION($P^1, \cdots, P^\ell, p, C^\ell, \mathfrak{C}^\ell$ where $\ell \in [L]$)

---

**Input:** $\ell$ copies $P^1, \cdots, P^\ell$ of the set of points that have been inserted but not deleted before the insertion of $p$. The sets $C^\ell$ and $\mathfrak{C}^\ell$ are the set of centers and clusters for the guess $\ell \in L$.

  1: **for** $\ell = 0$ to $L$ **do**
  2:      Insert $p$ into $P^\ell$
  3:      **if** there is an $\arg\min_{i \in [k]}$ for which $p \in B(c_i^\ell, 4(1+\varepsilon)^{\ell+1})$ **then**
  4:          Insert $p$ into cluster $\mathfrak{C}_i^\ell$
  5:          **if** $p \in B(c_i^\ell, 2(1+\varepsilon)^{\ell+1})$ **then**
  6:              $m_i^\ell \leftarrow m_i^\ell + 1$
  7:      **else**                     $\triangleright$ $p$ is not covered by the current set of clusters for the guess $(1+\varepsilon)^\ell$.
  8:          **if** there are $j < k$ clusters **then**
  9:              $C^\ell, Q^\ell, \mathfrak{C}^\ell \leftarrow C^\ell, Q^\ell, \mathfrak{C}^\ell \cup$ CLUSTERING($\{p\}, d, \varepsilon, \ell, j$)
10:          **else**
11:              Add $p$ to the remainder set $Q^\mathfrak{l}$
12: Let $\mathfrak{l} \in [L]$ be the smallest index for which $Q^\mathfrak{l}$ is empty
13: Return $C^\mathfrak{l}$ and $\mathfrak{C}^\mathfrak{l}$

---

For any point $p \in P$, let $c^*(p)$ denote the optimal center assigned to $p$, i.e., the center $c_i^*$ such that $p \in \mathfrak{C}_i^*$. Similarly, given any set of $k$ centers $C = \{c_1, \ldots, c_k\} \subset \mathbb{R}^d$, let $c(p) \in C$ denote the center to which $p$ is assigned in the clustering induced by $C$, i.e., the center $c_i$ such that $p \in \mathfrak{C}_i$. For the guess $(1+\varepsilon)^{\ell^*}$, the offline algorithm INIT($P, d, k, \varepsilon$) returns a set $C^{\ell^*} = \{c_1^{\ell^*}, \ldots, c_f^{\ell^*}\}$ of $f \leq k$ centers. Our goal is to maintain both the disjointedness and the closeness invariant.

Note that it immediately follows from Definition 4.1 that a $(\ell^*, 1)$-robust center of $\mathfrak{C}_i^*$ respects the closeness invariant.

**Lemma B.1.** *If both invariants hold for all clusters $\mathfrak{C}_i^{\ell^*}$ for $i \in [f]$, we have a $4(1+\varepsilon)$-approximate solution for the $k$-center clustering of the point set $P$.*

*Proof.* For the approximation factor, for every $i \leq f$ and for every point $p \in \mathfrak{C}_i^{\ell^*}$, the closeness invariant guarantees that

$$\text{dist}(p, c_i) \leq \text{dist}(p, c^*(p)) + \text{dist}(c^*(p), c_i^{\ell^*}) \leq OPT(P) + OPT(P) + \text{dist}(c_i^{\ell^*}, \mathfrak{C}_i^*),$$

is at most $4(1+\varepsilon)OPT(P)$. To show that all points are clustered, we construct a set $\{p_1, \ldots, p_k\}$ with each point at pairwise distance greater than $2OPT(P)$. Let $p_i \in \mathfrak{C}_i^{\ell^*}$ be the point closest to $c_i^{\ell^*}$, for which it holds that $\text{dist}(p_i, c_i^{\ell^*}) \leq 2(1+\varepsilon)OPT(P)$. For any $p_j$ with $j \neq i$, it holds that

---

**Algorithm 8** CLUSTERINGDELETION($P^1, \cdots, P^\ell, p, C^\ell, \mathfrak{C}^\ell$ where $\ell \in [L]$)

---

**Input:** $\ell$ copies $P^1, \cdots, P^\ell$ of the set of points that have been inserted but not deleted before the insertion of $p$. The sets $C^\ell$ and $\mathfrak{C}^\ell$ are the set of centers and clusters for the guess $\ell \in L$.

1: **for** $\ell = 0$ to $L$ **do**
2:     Delete $p$ from $P^\ell$
3:     **if** there exists an $i \in [k]$ for which $p \in \mathfrak{C}^i$ **then**
4:         Delete $p$ from cluster $\mathfrak{C}_i^\ell$
5:         **if** $p \in B(c_i^\ell, 2(1+\varepsilon)^{\ell+1})$ **then**
6:             $m_i^\ell \leftarrow m_i^\ell - 1$
7:         **if** $m_i^\ell$ equals zero **then**
8:             $C^\ell, Q^\ell, \mathfrak{C}_i^\ell \leftarrow$ CLUSTERING($\cup_{j=i}^k \mathfrak{C}_j^\ell, d, \varepsilon, \ell, i$)
9:     **else**         ▷ $p \in Q^\ell$, i.e., is not covered by the current set of clusters for the guess $(1+\varepsilon)^\ell$.
10:         $Q^\ell \leftarrow Q^\ell \setminus p$         ▷ Delete $p$ from the remainder set $Q^\ell$
11:         Let $i \leftarrow |C^\ell|$ be the number of clusters stored for the guess $(1+\varepsilon)^\ell$
12:         $C^\ell, Q^\ell, \mathfrak{C}_i^\ell \leftarrow$ CLUSTERING($Q^\ell, d, \varepsilon, \ell, i+1$)
13: Let $\mathfrak{l} \in [L]$ be the smallest index for which $Q^j$ is empty
14: Return $C^{\mathfrak{l}}$ and $\mathfrak{C}^{\mathfrak{l}}$

---

$\text{dist}(p_j, c_i^{\ell^*}) \geq 4(1+\varepsilon)OPT(P)$ by the disjointedness property. Therefore,

$\text{dist}(p_i, p_j) \geq \text{dist}(p_j, c_i^{\ell^*}) - \text{dist}(p_i, c_i^{\ell^*}) \geq 4(1+\varepsilon)OPT(P) - 2(1+\varepsilon)OPT(P) > 2OPT(P).$

This means that each $p_i$ belongs to a different optimal cluster $\mathfrak{C}_i^*$. Hence, all optimal clusters are covered.      $\square$

Thus, to prove correctness, it is enough to show that our invariants hold throughout. We will need a few auxiliary definitions and lemmas.

The centers $C^{\ell^*} = \{c_1^{\ell^*}, \ldots, c_f^{\ell^*}\}$ are ordered according to the iteration in which they were found by Algorithm 6; that is, $c_1^{\ell^*}$ is computed first, followed by $c_2^{\ell^*}$, and so on up to $c_f^{\ell^*}$. Observe that the centers $c_1^{\ell^*}, \ldots, c_f^{\ell^*}$ are obtained from sampled points $a_1^{\ell^*}, \ldots, a_f^{\ell^*}$ that are sampled from optimal clusters. Suppose we order the optimal clusters so that $a_1^{\ell^*}$ is sampled from the optimal cluster $\mathfrak{C}_1^*$, and so on up to $a_f^{\ell^*}$ that is sampled from the optimal cluster $\mathfrak{C}_f^*$. Observe that if $f < k$, there are optimal clusters from which we have not sampled any point. In particular, we place these clusters arbitrarily after the last optimal cluster $\mathfrak{C}_f^*$ for which we have the last sampled point $a_f^{\ell^*}$.

We first prove that the Algorithm INIT (Lemma B.4) satisfies both invariants and that it constructs centers that are $(\ell^*, \frac{1}{2}|S_i^\ell \cap \mathfrak{C}_i^*|)$-robust. Consider an iteration $i \in [f]$ of Alg. 6. We now prove that $a_i^{\ell^*}$, if sampled, is a robust center for $\mathfrak{C}_i^*$.

**Lemma B.2.** *At iteration $i$, let $a_i^{\ell^*}$ be the point that is sampled from the optimal cluster $\mathfrak{C}_i^*$. The point $a_i^{\ell^*}$ is an $(\ell^*, |S_i^\ell \cap \mathfrak{C}_i^*|)$-robust center. That is, it satisfies $\mathfrak{C}_i^* \subset B(a_i^{\ell^*}, 2(1+\varepsilon)^{\ell^*+1})$.*

*Proof.* Since $a_i^{\ell^*} \in \mathfrak{C}_i^*$ and the cluster $\mathfrak{C}_i^*$ that is centered at the optimal center $c_i^*$ has radius $OPT(P^{\ell^*})$, we conclude that $\text{dist}(q, a_i^{\ell^*}) \leq \text{dist}(q, c_i^*) + \text{dist}(c_i^*, a_i^{\ell^*}) \leq 2OPT(P^{\ell^*}) \leq 2(1+\varepsilon)^{\ell^*+1}$ for any point $q \in \mathfrak{C}_i^*$. Thus, the claim of this lemma holds.      $\square$

Let us define inactive points $\mathcal{IN}_{i-1}^{\ell^*} = \cup_{t=1}^{i-1}\mathfrak{C}_t^{\ell^*}$ and active point $\mathcal{AC}_{i-1}^{\ell^*} = P^{\ell^*} \setminus \mathcal{IN}_{i-1}^{\ell^*}$. In the first iteration $i = 1$, we have $\mathcal{IN}_0^{\ell^*} = \emptyset$ and $\mathcal{AC}_0^{\ell^*} = P^{\ell^*}$. We say that an optimal cluster $\mathfrak{C}_j^*$ is *active* at iteration $i$ if $\mathfrak{C}_j^* \not\subset \mathcal{IN}_{i-1}^{\ell^*}$; otherwise, $\mathfrak{C}_j^*$ is an *inactive* cluster. Next, we prove that if there is a center sufficiently robust, then we expect to be able to identify it through our sampling process.

**Lemma B.3.** *Consider a timestep $t \in [n]$ and an iteration $i \in [f]$ of Algorithm CLUSTERING. Define $N_{i-1} = \mathcal{AC}_{i-1}^{\ell^*}$ and $N^j = |\mathfrak{C}_j^* \cap \mathcal{AC}_{i-1}^{\ell^*}|$. With failure probability at most $\delta$, we identify a $(\ell^*, \frac{N_{i-1}}{4(k-i)})$-robust center successfully through our sampling process. Moreover, the total computational cost of sampling and counting is bounded by $\tilde{O}(\min\{(k-i)^2, \sqrt{(k-i)N_{i-1}}\})$.*

*Proof.* We consider two possible cases based on the value of $k - i$. If $(k - i) \leq N_{i-1}^{2/3}$, set $|S_i^\ell| = C(k-i) \log L \log(n/\delta)$, for some sufficiently large constant $C > 0$. Then, using a simple Chernoff bound argument, it is easy to verify that with probability at least $1 - \frac{\delta}{\log_{(1+\varepsilon)}(\rho) \cdot n^4}$, our sampling process satisfies the following two conditions:

- If an optimal active cluster is small, i.e., $N^j < \frac{N_{i-1}}{4(k-i)}$, then $|S_i^\ell \cap \mathfrak{C}_j^*| < \frac{|S_i^\ell|}{3(k-i)}$.

- If an optimal active cluster is large, i.e., $N^j \geq \frac{N_{i-1}}{2(k-i)}$, then $|S_i^\ell \cap \mathfrak{C}_j^*| \geq \frac{|S_i^\ell|}{3(k-i)}$.

Since there exists an optimal cluster with at least $\frac{N_{i-1}}{(k-i)}$ unclustered points, if the above conditions hold, then the cluster with the highest count provides a sample point that is $(\ell^*, \frac{N_{i-1}}{4(k-i)})$-robust. The cost of this process is bounded by $\tilde{O}(|S_i^\ell|^2) = \tilde{O}((k-i)^2) = \tilde{O}(\sqrt{k}N_{i-1})$, by the assumption on $k$.

In the other case, $(k-i) \geq N_{i-1}^{2/3}$, we use a lower sampling probability and show that we can still find a robust center. Let us set $|S_i^\ell| = C\sqrt{(k-i)} \log L \log(n/\delta)$. We compute exactly for each $q \in S_i^\ell$ the value of $|B(q, 2(1+\varepsilon)^{\ell^*+1}) \cap Y|$ and select the point with the highest count. To show that we find a robust center, we proceed with a case distinction. If there are at least $\sqrt{(k-i)}$ clusters of size at least $N_{i_1}/4(k-i)$, through a similar Chernoff bound argument, we are ensured that there is a sample point from one such clusters which provides the desired robust center. Otherwise, the smallest $(k-i) - \sqrt{(k-i)}$ clusters contain at most $N_{i-1}/4$ points out of $N_{i-1}$. This means that the largest cluster has at least $N_{i-1}/2\sqrt{(k-i)}$ points, and, through an analogous Chernoff bound argument, we can prove that one such point will be clustered. The cost of this process is bounded by $\tilde{O}(|S_i^\ell|N_{i-1}) = \tilde{O}(\sqrt{(k-i)}N_{i-1}) = \tilde{O}((k-i)^2)$.

By a union bound over the at most $k$ clusters and $k$ iterations per guess $\ell \in [L]$, the probability that none of them fail, is at least $1 - \frac{k^2 \cdot L}{\log_{(1+\varepsilon)}(\rho) \cdot n^4} \geq 1 - \frac{\delta}{2n^2}$, since $k \leq n$. Further, by a final union bound on the number of timesteps $t \in [n]$, the probability of failure is at most $\delta$. Finally, we remark that the algorithm does not need to know $n$ in advance; the same guarantee can be obtained by replacing $n$ with $t$, as in the diameter case. We also note that the failure of this randomized process cannot be affected by an adversary. $\qquad\square$

**Lemma B.4.** *After we invoke Algorithm* INIT$(P, d, k, \varepsilon)$, *each center $c_i^{\ell^*}$ is $(\ell^*, |\mathcal{AC}_{i-1}^{\ell^*}|/4(k-i))$-robust and both of our invariants hold.*

*Proof.* Let us consider the iterations of Algorithm CLUSTERING for the guess $(1+\varepsilon)^{\ell^*}$ such that $(1+\varepsilon)^{\ell^*} \leq OPT(P) < (1+\varepsilon)^{\ell^*+1}$. We proceed by induction.

For $i = 1$ and $\mathcal{AC}_0^{\ell^*} = P^{\ell^*}$, by Lemma B.3, we find an $(\ell^*, n/4k)$-robust center, which we can label $\mathfrak{C}_1^*$ without loss of generality. Let $\mathfrak{C}_1^{\ell^*}$ denote the set of points in $B(c_1^{\ell^*}, 4(1+\varepsilon)^{\ell^*+1}) \cap \mathcal{AC}_0^{\ell^*}$. According to Lemma B.2, we have $(\mathfrak{C}_1^* \cap \mathcal{AC}_0^{\ell^*}) = \mathfrak{C}_1^* \subseteq \mathfrak{C}_1^{\ell^*}$. Therefore, the optimal cluster $\mathfrak{C}_1^*$, which was active before iteration $i$, becomes inactive in this iteration. The disjointedness invariant is vacuous for $i = 1$.

For $i > 1$, assume that both invariants holds for the clusters found so far. Since all points within distance $4(1+\varepsilon)^{\ell^*+1}$ from the previous centers have been removed, the disjointedness invariant continues to hold. There must exist one active optimal cluster with at least $\frac{|\mathcal{AC}_{i-1}^{\ell^*}|}{4(k-i)}$ points. (If no such cluster exists, the process terminates and nothing more needs to be proven.) By Lemma B.3, we obtain an $(\ell^*, |\mathcal{AC}_{i-1}^{\ell^*}|/4(k-i)$-robust center, which we can label by $\mathfrak{C}_i^*$. Define $\mathfrak{C}_i^{\ell^*}$ as $B(c_i^{\ell^*}, 4(1+\varepsilon)^{\ell^*+1}) \cap \mathcal{AC}_{i-1}^{\ell^*}$. Then, by Lemma B.2, we have $(\mathfrak{C}_i^* \cap \mathcal{AC}_{i-1}^{\ell^*}) \subseteq \mathfrak{C}_i^{\ell^*}$. Thus, the optimal cluster $\mathfrak{C}_i^*$, which was active before iteration $i$, becomes inactive at this iteration. The disjointedness invariant also holds since we filter all points within the desired radius. $\qquad\square$

Next, we prove that our invariants are respected at any time $t$, upon insertion or deletion of an arbitrary point $p$ after invoking Algorithms CLUSTERINGINSERTION or CLUSTERINGDELETION, respectively.

**Lemma B.5.** *Suppose both of our invariants hold before time $t$. If an arbitrary point $p$ is inserted or deleted at time $t$, both invariants will also hold after invoking Algorithm* CLUSTERINGINSERTION *or* CLUSTERINGDELETION.

*Proof.* Let us consider the disjointedness invariant first. The deletion of a point cannot violate it. For the case of insertion, we assign the newly inserted point to the lexicographically smallest cluster that contains it (if any), which ensures disjointedness.

For the closeness invariant, the insertion of a point cannot violate it. In the case of a deletion of a point at distance at most $2(1 + \varepsilon)OPT(P)$ from $c_i^\ell$, the counter $m_i^\ell$ is updated and, if zero is reached, we recompute subsequent clusters using INIT. Since the invariant for clusters $i' < i$ are not affected, the execution of INIT will establish the invariants again for cluster $i$ and subsequent ones. $\square$

## B.5 Runtime Analysis

**Lemma B.6.** *The running time of Algorithm* CLUSTERING$(P, d, \varepsilon, \ell, i)$ *is* $\tilde{O}(d(k-i)^{1.5}|\mathcal{AC}_i^\ell|)^3$.

*Proof.* The algorithm consists of $(k - i)$ iterations for a fixed guess $(1 + \varepsilon)^\ell$. Each iteration $j = i, \ldots, k$ takes $\tilde{O}(d\sqrt{k-j}\mathcal{AC}_j^\ell)$ by Lemma B.3. Therefore, the total computation is bounded by $\tilde{O}(d(k-i)^{1.5}|\mathcal{AC}_i^\ell|)$. $\square$

**Lemma B.7.** *Consider a sequence $S$ of $n$ possibly adversarial updates, and let $P_t$ be the point set obtained after applying the first $t$ updates. The amortized update time per operation is* $\tilde{O}(dk^{2.5})$.

*Proof.* The total number of guesses $L$ incurs only an extra $\tilde{O}(1)$ factor, so we can consider an arbitrary $\ell$. Note that insertions are handled in $\tilde{O}(dk)$ time and that if at any time the size of the point set is $\tilde{O}(k^{1.25})$, then we can obtain a 2-approximation by computing all pairwise distances. When a point $p$ is deleted, there are three possible cases.

If $p$ belongs to some cluster $i$ and $p$ is within distance $2(1 + \varepsilon)^{\ell+1}$ from its center, then $m_i^\ell$ will be decreased by one. If $m_i^\ell > 1$, then the update time is $\tilde{O}(dk)$. If $m_i^\ell = 1$, then the $i$-th cluster along with subsequent ones need to be reconstructed. Consider first cluster $i = 1$. We prove the amortized bound by induction on the number of robust centers $w$ that are computed throughout the algorithm for cluster $i = 1$. For $w = 1$, we obtain an $(\ell, |P_t|/4k)$ centerpoint in $\tilde{O}(d\sqrt{k}|P_t|)$ time as the first robust center and proceed to compute the remaining $k - 1$ clusters. The total computation cost of $\tilde{O}(dk^{1.5}|P_t|)$ can be amortized over the first $|P_t|$ insertions, resulting in $\tilde{O}(dk^{1.5})$ charge. For the inductive step, we amortize the computation of the $(w + 1)$-th robust center at time $t_{w+1}$ over two operations: (1) the decreases that $m_i^\ell$ experienced and (2) the updates that occurred between time $t_{w+1}$ and time $t_w$, when the computation of the $w$-th robust center occurred. Since the computation of the $w$-th centerpoint is amortized over updates prior to $t_w$, it does not affect the amortization on subsequent updates. The total computation to amortize is $\tilde{O}(dk^{1.5}|P_{t_{w+1}}|)$. Now, recall that $t_{w+1} - t_w \geq |P_t|/4k$ by the robust center property, and observe that

$$\frac{|P_{t_{w+1}}|}{t_{w+1} - t_w} \leq \frac{|P_{t_w}|}{t_{w+1} - t_w} + \frac{t_{w+1} - t_w}{t_{w+1} - t_w} = \tilde{O}(k) + \tilde{O}(1).$$

This means that the points who caused $m_i^\ell$ to decrement receive a charge of $\tilde{O}(dk^{2.5})$ from cluster $i = 1$, while all other points (which did not cause $m_i^\ell$ to drop) receive a charge of $\tilde{O}(d)$ from cluster $i = 1$.

Consider now $i > 1$. The induction proceeds in a similar way. The amortization of the first robust center for the $i$-th cluster is straightforward. For the inductive step, we again amortize the computation of the $(w + 1)$-th robust center of cluster $j > 1$ at time $t_{w+1}$ over the decrements as well as updates that occurred between time $t_{w+1}$ and time $t_w$. It is worth noting that deletions of points from clusters

---

[3]$\tilde{O}$ hides the polynomial factors of $\varepsilon^{-1}, \log(n), \log \varphi^{-1}$.

$j > i$ cannot affect $m_i^\ell$ at all, whereas deletions from clusters $j < i$ can only positively affect $m_i^\ell$ since they might reset it without additional amortization charges. Let $N_{i-1}^{t_w}$ be the number of active points at time $t_w$ at the beginning of iteration $i$. The total computation to amortize is $\tilde{O}(d(k-i)^{1.5} N_{i-1}^{t_{w+1}})$. Now, recall that $t_{w+1} - t_w \geq N_{i-1}^{t_w}/4(k-i)$ by the robust center property, and observe that

$$\frac{N_{w-1}^{t_{w+1}}}{t_{w+1} - t_w} \leq \frac{N^{t_w}}{t_{w+1} - t_w} + \frac{t_{w+1} - t_w}{t_{w+1} - t_w} = \tilde{O}(k-i) + \tilde{O}(1).$$

Again, the points who caused $m_i^\ell$ to decrement receive a charge of $\tilde{O}(dk^{2.5})$ from some cluster $i > 1$, while all other points (which did not cause $m_i^\ell$ to drop) receive a charge of $\tilde{O}(d)$ from any other cluster $i > 1$. Since each point belongs to a single cluster, it receives one charge of $\tilde{O}(dk^{2.5})$ from its own cluster, at most $k - 1$ charges of $\tilde{O}(d)$ from other clusters, and at most $k$ charges of $\tilde{O}(dk^{1.5})$ when a cluster is created for the first time, proving the claimed amortized update time.

It remains to discuss the case in which $p$ does not belong to any cluster and it is deleted from the remainder set. If the invocation of CLUSTERING creates a new robust center for some $i$, we can apply induction as before. If the new cluster was created for the first time, we can charge the insertions of those points. If the cluster previously had a robust center, we can use the same argument as above. Finally, if the guess is too low and no cluster can be created, there is a one-time charge of $O(k^2)$ for running Lemma B.3, after which the computation is halted. $\qquad\square$

