# OpenReview forum: "Dynamic Diameter in High-Dimensions against Adaptive Adversary and Beyond"
_NeurIPS.cc/2025/Conference — NeurIPS 2025 poster_

### Official Review · Reviewer_SCZU · 2025-06-29

**Clarity:** 3
**Significance:** 3
**Originality:** 3
**Rating:** 5
**Confidence:** 2

**Summary:**

The paper tackles two classical geometric problems—maintaining the diameter and $k$-center clustering—over fully dynamic point sets in high-dimensional Euclidean space, under the strongest “randomness-adaptive” adversary model.

The main contributions are
1. a 2-approximation algorithm for dynamic diameter (and minimum enclosing ball) with worst-case update time $O\bigl(d^{5}\log^{1.5}d\log^{1.5}(n/\delta)\bigr)$, and
1. a $(4+\varepsilon)$-approximation for fully-dynamic $k$-center with amortized update time $k^{2.5}d\cdot \mathrm{poly}(\varepsilon^{-1},\log n)$, improving the previous $k^6\cdot \mathrm{poly}(d, \varepsilon^{-1}, \log n)$ bound.

Both results hinge on a novel use of $\varepsilon$-robust centerpoints (Tukey-depth ≥ $\Omega(1/d^2)$) to pick "representative" anchors that survive many deletions.
The diameter algorithm also employs a careful de-amortization step.

**Questions:**

N/A

**Ethical Concerns:**

["NO or VERY MINOR ethics concerns only"]

**Final Justification:**

I believe the improvement in the results are significant and will be a nice contribution to NeurIPS.

**Limitations:**

yes

**Quality:**

4

**Strengths And Weaknesses:**

### Strengths
1. Fully dynamic algorithms for the diameter problem against adaptive adversaries with any non-trivial
approximation and worst-case $\mathrm{poly}(d \log n)$ update time were previously not known.
1. The improvement in update time for fully dynamic, adaptive $k$-center seems significant.
1. The robust centerpoint idea is clean and may be applicable in other settings.

---

### Weaknesses
1. The $d^5$ update time may be impractical for high-dimensional settings.

---

> ### Author Rebuttal · Authors · 2025-07-31
>
> Thank you for your valuable feedback and for the time spent reviewing our paper. We are pleased that you recognize the significance of our fully dynamic diameter algorithm and the improved update time for k-center. We're also glad that you found the robust centerpoint idea clean and broadly applicable.

---

> > ### Comment · Reviewer_SCZU · 2025-08-06
> >
> > Thanks for the reply. I look forward to reading the final version of your work and will keep my score.

---

> ### Comment · Area_Chair_rXUG · 2025-08-05
> **Please respond to author rebuttals**
>
> Dear Reviewer SCZU,
>
> Please note that only flagging "mandatory ack" is not sufficient, and please actively participate in the discussion with the authors. This could be short (for example, you think your comments are fully addressed), but you should anyway respond.
>
> Regards,
>
> AC

---

### Official Review · Reviewer_8vVV · 2025-07-01

**Clarity:** 4
**Significance:** 3
**Originality:** 3
**Rating:** 5
**Confidence:** 4

**Summary:**

The paper presents two fully dynamic algorithms:
* dynamic maintenance of a 2-approximation to diameter in O(d^5 log^3 n) worst case update time
* dynamic maintenance of (4+epsilon)-approximate k-center in O(k^2.5 d poly(epsilon^-1, log n)) amortized update time
Here, d is the dimension of the underlying Euclidean space, and n is the current number of points.

Both results hold against a strong notion of adaptive adversaries, called randomness-adaptive adversary: the adversary has access to the algorithm, all random choices and outputs of the algorithm up to that point.

The diameter result relies on finding an epsilon-robust representative point of pointset P - intuitively, a point x such that dist(x, P) stays a 2 approximation to diam(P) even after any epsilon*n changes to set P. Finding such a point guarantees that we do not need to recompute the representative (and hence the diameter) of the set too often, and that the algorithm is also independent of the exact updates chosen by the adversary. The computation of the representative point is reduced to a black-box algorithm for finding an alpha-centerpoint (for alpha=1/poly d). After epsilon*n deletions to P, we simply recompute the representative point from scratch. The deamortization of this algorithm uses a standard technique by Motwani and Raghavan (1999).

The k-center algorithm does the following for each “guess” (1+epsilon)^l of the size of the optimal solution OPT(P). In the first round, we sample some points from P, and choose a point out of these that has most of the remaining points at distance at most 2(1+epsilon)^l. The chosen center, together with its closest points, is then deleted from P, and we recurse on the remaining points. Intuitively, if our “guess” of OPT(P) was good, we only have k rounds, and hence choose k centers.

**Questions:**

Is there a possibility of deamortizing the k-center algorithm? I assume that the recursive nature of the algorithm might make this hard, but maybe one can make these rounds of choosing the centers less dependent on each other.

Some minor comments about the presentation:
Table 1 shows dependencies of d, while Table 2 doesn’t, making it a bit inconsistent.
Line 65: these performance measures also apply to the diameter problem.
Line 128: citation format issue
Line 201: who see -> who sees
Line 233: should be P\in R^d

**Ethical Concerns:**

["NO or VERY MINOR ethics concerns only"]

**Final Justification:**

The paper studies a timely problem, achieves relevant results, and it's very well written. I recommend acceptance and look forward to reading the final version.

**Limitations:**

yes

**Quality:**

3

**Strengths And Weaknesses:**

Strengths:
* The paper is well written, easy to follow, and provides good intuition of the techniques used. Even simple results are proven in detail in the appendix, making the whole paper self-contained.
* The literature overview is comprehensive, and the summary of techniques in other papers clearly highlights the contributions of this work.
* The results hold for a strong notion of adaptive adversaries: the adversary has access to the algorithm, all random choices, and outputs of the algorithm up to that point.

Weaknesses:
* While the diameter result uses some novel techniques, I think paying a large polynomial dependency on dimension just for getting worst-case guarantees feels like overkill to me.
* The k-center result is less novel technically, but shows a clear improvement in the dependency on the number of centers. To me, it would be interesting to get such a result for the 2+eps approx regime, which is known to be tight for this problem.
Some minor comments (see Questions section).

---

> ### Author Rebuttal · Authors · 2025-07-31
>
> Thank you for your thorough feedback and for the time spent reviewing our paper. We appreciate your positive review and the helpful minor comments on the presentation. We answer your question below.
>
> > Is there a possibility of deamortizing the k-center algorithm? I assume that the recursive nature of the algorithm might make this hard, but maybe one can make these rounds of choosing the centers less dependent on each other.
>
> Thanks for this very good question. It seems that the recursive nature of the algorithm mostly plays a role in the update time of the algorithm, causing an extra O(k) factor. The problem that we had with the deamortization was that we could not deamortize different guesses of OPT, as the ‘right’ guess might change multiple times during the course of the deamortization.

---

> > ### Comment · Reviewer_8vVV · 2025-08-06
> >
> > Thanks for your reply and for sharing your insights regarding the challenges of achieving worst-case bounds with your algorithm.
> >
> > I don't have any further questions. I retain my positive score, and look forward to reading the final version of your paper.

---

> ### Comment · Area_Chair_rXUG · 2025-08-05
> **Please respond to author rebuttals**
>
> Dear Reviewer 8vVV,
>
> Please note that only flagging "mandatory ack" is not sufficient, and please actively participate in the discussion with the authors. This could be short (for example, you think your comments are fully addressed), but you should anyway respond.
>
> Regards,
>
> AC

---

### Official Review · Reviewer_EhRa · 2025-07-02

**Clarity:** 3
**Significance:** 3
**Originality:** 3
**Rating:** 4
**Confidence:** 3

**Summary:**

The paper is about fully dynamic (allowing insertions & deletions) constant factor approximation algorithms for the diameter and k-center clustering of the input.

Moreover the algorithms are robust against adversaries that have full knowledge of the past decisions and random bits used by the algorithm, which is argued to be the strongest model that admits constant factor approximations.

The paper combines in a clever way, simple online algorithm with recent advances in computational geometry and with classical robust statistics.

the algorithm for maintaining the diameter is super simple and uses in a clever way the robustness of so called Tukey centerpoints, i.e., points that lie *deeply* in the interior of the point set:

Any point (not necessarily in the input) in the convex hull (CH) of the current set of points is a 2-approximation for the diameter (or 1-center). The computed centerpoint lies deeply in the CH and has the property that at least d^2 many points need to be removed before it may fall out of the CH.

A caveat is that such points are relatively expensive to compute, in roughly d^7. But once computed, they may not fall out of the CH on insertions and since they remain valid 2-approximations for at least d^2 deletions. Thus, the amortized time is only about d^5. The calculations steps are additionally cleverly distributed over all steps to additionally de-amortize the algorithm.

The contributions are well compared to prior work in two tables covering dynamic k-center and diameter problems, respectively.

**Questions:**

* to what extent do you think can the approximation factor be improved? Any comments?

* to what extent do you think can the update time be improved? Why do you state O(k^6) as being "prohibitively large" even for small $k$ for k-center while O(d^5) is good enough for the diameter?

* for k-center, any ideas on reaching linear k dependence as in prior work while dealing with an adaptive adversary?

* I'd like to see a more formal definition of the Definition 2.4 (Tukey Depth). The current one reads more like an informal description.

* I'd like to see a citation of Tukey's original work. Some more or less recent contributions from the computational geometry community are present, but the classical statistical work should also be cited. Especially for such an important notion named after its inventor.

**Ethical Concerns:**

["NO or VERY MINOR ethics concerns only"]

**Final Justification:**

I'd like to see this paper published at the current NeurIPS due to its originality in combining results from different fields. However, I also have slight reservations, because it is unclear whether the presented techniques can be improved to below the (folklore) 2-approximation  or to significantly better d-dependence. For this reason, I have not increased my rating any further.

**Limitations:**

yes

**Quality:**

3

**Strengths And Weaknesses:**

Strengths:
the paper is well written, and contains a good presentation of related work and comparisons to prior results in similar settings.

simplicity of the algorithms

clever combination of algorithmic concepts with computational geometry and basic robust statistics.


Weakness:
The update time is rather high, and even though previous algorithms had other caveats, and in particular did not work against adaptive adversaries, they had achieved $1+\epsilon$ or $\sqrt{2}$ approximation which is better than the presented 2-approx.

---

> ### Author Rebuttal · Authors · 2025-07-31
>
> Thank you for your valuable feedback and for the time spent reviewing our paper. We are glad that the simplicity of our algorithms is valued and that you find the combination of computational geometry with basic robust statistics clever. We address the questions below.
>
> > To what extent do you think the approximation factor can be improved? Any comments?
>
> We believe this to be an excellent direction for future work. While we thought about improving the factor using ideas from Chan and Pathak, we were unable to adopt their techniques to our setting. On a high level, the key problem is that the algorithm relies on obtaining a set of anchor points which are all far from each other, whereas our techniques intuitively try to do the opposite since we look at the centroid of the datapoints. Currently, it is not clear whether these ideas can be merged and we suspect that improving the approximation factor would require an approach that is distinct from the prior work in the non-adaptive setting.
>
> > to what extent do you think the update time can be improved?
>
> Since a main bottleneck in the update time is the running time of the algorithm for finding centerpoints, we suspect that the poly(d) factor can be improved. Obtaining a time that does not depend on the complexity of finding a centerpoint is likely much more challenging and would require more complex techniques.
>
> > Why do you state O(k^6) as being "prohibitively large" even for small k for k-center while O(d^5) is good enough for the diameter?
>
> The value of $d$ is generally assumed to be polylogarithmic in $n$ while no assumption is made on $k$. On a high level this is because at most log(n) dimensions are generally sufficient to capture the geometry of a dataset. Indeed, the Johnson–Lindenstrauss lemma shows any dataset of points in $R^d$ can be projected into a dataset in $R^{d’}$ where $d’ \in \Theta(\log(n))$ while approximately preserving distances.
>
> > for k-center, any ideas on reaching linear k dependence as in prior work while dealing with an adaptive adversary?
>
> While we currently do not have any ideas for either obtaining a linear-time algorithm or establishing hardness, we suspect that obtaining such an algorithm would require substantially new techniques and may in fact be impossible. Indeed, the prior work of Bateni et al. for adaptive adversaries has an approximation ratio of $O(\min\{k, \log(n/k)/\log\log(n) \})$. This suggests that obtaining any constant factor approximation with linear update time is a challenging problem and would likely require substantially new techniques.
>
> > I'd like to see a more formal definition of the Definition 2.4 (Tukey Depth) …
>
> We appreciate the suggestion to use a more formal definition for the Tukey Depth and to include a citation to Tukey’s original work. We provide a more formal definition below which we have included in the paper and have added the appropriate citation [1].
>
> Given a point set $P = \{p_1, \dots, p_n\} \subset \mathbb{R}^d$, the **Tukey depth** of a point $x \in \mathbb{R}^d$ with respect to $P$ is defined as:
> $
> d_P(x) = \min_{\|v\|=1} \sum_{i=1}^n \mathbf{1} [v^\top (p_i - x) \ge 0 ]
> $
>
> [1] Tukey, John W. "Mathematics and the picturing of data." In Proceedings of the international congress of mathematicians, vol. 2, pp. 523-531. 1975.
>
>
> Finally, if you feel that our response has adequately addressed your major concerns, please, consider adjusting your score accordingly.

---

> > ### Author Response · Authors · 2025-08-03
> >
> > Dear Reviewer,
> >
> > We would like to kindly check whether you have any remaining questions or concerns. We would be happy to provide further clarification. If not, we would be grateful if you could consider updating your score in light of the rebuttal.

---

> > ### Comment · Reviewer_EhRa · 2025-08-04
> > **Thank you for the clarifications**
> >
> > I would like to thank the authors for their clarifications in response to my review. I have already acknowledged the submission with mostly positive feedback and a positive rating. I am currently not inclined to raise the score further, as I am not sufficiently impressed by the resulting approximation and runtime. The author's response indicates that their current approaches are to some extent limited and do not allow to go very far beyond the novel and clever combination* of existing techniques.
> >
> > *which I like very much!

---

> ### Comment · Area_Chair_rXUG · 2025-08-04
> **Respond to author rebuttal**
>
> Dear Reviewer EhRa,
>
> Would you please check if the authors address your major concerns, and if you have further comments? A minimal response would be to submit your “Mandatory Acknowledgement”.
>
> Regards,
>
> AC

---

### Official Review · Reviewer_LwQb · 2025-07-03

**Clarity:** 3
**Significance:** 3
**Originality:** 3
**Rating:** 4
**Confidence:** 4

**Summary:**

In this paper, the authors study the problem of maintaining the diameter and k-center clustering under dynamic updates (addition/deletion of points) under the presence of an adaptive adversary. The authors propose fully dynamic algorithms that achieve a 2-approximation for the diameter problem and a $(4 + \epsilon)$-approximation for the k-center clustering problem. A simple 2-approximation dynamic algorithm picks a random anchor point $p \in P$ and maintains its furthest neighbor throughout updates, remains the best known in terms of both approximation and update time. In this work, the authors provide a significant improvement, by designing a 2-approximation algorithm with worst case update of $O(d^5)$ under adaptive adversaries. Their work introduces novel techniques such as robust representatives based on center points and de-amortization strategies to ensure worst-case update time guarantees.

**Questions:**

It would have been great, if the authors highlighted the intution behind their novel sampling to improve update time.

**Ethical Concerns:**

["NO or VERY MINOR ethics concerns only"]

**Final Justification:**

I'm happy with rebuttal response. As I asked for clarifying questions, I am keeping my score.

**Quality:**

3

**Strengths And Weaknesses:**

**Strengths**:

The theoretical claims are strong (improvements over both the approximation guarantee and worst-case update time) and the algorithm relies on highly non-trivial techniques.

The guarantees hold against an adaptive adversary that also has access to randomness of the algorithm, which is a significantly harder case.

**Weaknesses**:

The authors heavily rely on a prior work to find the center point. Essentially these center points serve as the anchor points (from folklore).

The authors explain the role of adaptive adversaries, motivating it. Such an adversary model is common in the literature, and the writing could be improved to focus more on the overview of technical challenges.

---

> ### Author Rebuttal · Authors · 2025-07-31
>
> Thank you for your valuable feedback and for the time spent reviewing our paper. We are encouraged by your positive comments, in particular that our techniques are highly non-trivial and that the model considered is significantly harder. We address the main concerns and questions below.
>
> > It would have been great, if the authors highlighted the intuition behind their novel sampling to improve update time.
>
> Thank you for suggesting to add the intuition behind our novel sampling technique. Our sampling mechanism runs in time $\tilde{O}(\min(k^2, \sqrt{k}n))$ (see Lemma B.3). This enables a total running time of $\tilde{O}(dk^{1.5}n)$ for the offline clustering algorithm (see Lemma B.6), and an amortized update time of $\tilde{O}(dk^{2.5})$ for our dynamic algorithm under adversarial updates (see Lemma B.7). We explain the intuition behind this novel sampling method in the updated version of the paper. For completeness, we provide a summary here.
>
> Recall that the robustness (see Definition 4.1) of a point $x$ is defined as the fraction of points in the dataset $P$ that lie within a ball centered at $x$ of a given radius. A higher robustness implies that $x$ is a stronger candidate for a center, as it can serve effectively over a longer sequence of dynamic updates.
>
> To estimate the robustness of sampled points efficiently, we take into account that there are $k$ clusters. We sample $\tilde{O}(k)$ points and estimate the robustness of each by measuring the fraction of other sampled points lying within a ball of radius $2\cdot OPT$ around it (where $OPT$ denotes the optimal clustering cost). This allows us to identify a robust point—i.e., one that covers the most sampled points—in time $\tilde{\Theta}(k^2)$. Note that since $OPT$ is not known in advance, we use a logarithmic number of guesses, one of which is guaranteed to be within a constant factor of the true $OPT$.
>
> This robustness estimation mechanism is efficient for our dynamic algorithm when the number of clusters $k$ is relatively small, specifically when $k \leq n^{2/3}$, yielding a total running time of $\tilde{O}(k^2)$.
>
> When $k$ exceeds this threshold, we employ a different mechanism. In this case, we reduce the sample size to $\tilde{O}(\sqrt{k})$ and modify our robustness estimation strategy. Instead of testing the fraction of sampled points within radius $2\cdot OPT$ around each sample point, we test the fraction of original points in $P$ that lie within such a ball.
>
> There are two cases:
>
> Case 1: The largest $\sqrt{k}$ clusters each have size at least $n/k$.
> In this case, with high probability, each of these large clusters will be represented in the sample. Estimating robustness based on coverage of original points will allow us to identify robust centers from these clusters.
>
> Case 2: The remaining $k - \sqrt{k}$ clusters collectively contain at most $n/t$ points, for some constant $t$ (e.g., $t = 4$).
> Since the large clusters contain at least $(1 - 1/t)n$ points, by the pigeonhole principle, there exists a cluster with at least $\frac{n}{4\sqrt{k}}$ points. Sampling $\tilde{O}(\sqrt{k})$ points uniformly at random ensures, with high probability, that at least one point from this large cluster is included. We can then test its robustness against the full dataset in time $\tilde{O}(\sqrt{k}n)$.
>
>
> > The authors heavily rely on a prior work to find the center point. Essentially these center points serve as the anchor points (from folklore).
>
> There are two novelties in our dynamic algorithm for the diameter problem. The first is the connection we made between dynamic algorithms against adaptive adversaries and the concept of a center point.
> The concept of a center point is known in computational geometry and machine learning; however, its role in making dynamic algorithms resilient to adaptive adversaries is new and has not been explored before. We think that bridging a connection between two different lines of research would be as interesting as developing new techniques for one line of research. It in fact shows that tools classical in one area may help improve achievements in other areas. The second novelty is the set of lemmas that prove that center points can be maintained in a dynamic setting. This includes Lemma 2.10 and 20.9, which show an α-center point of a point set lies inside the convex hull of that set, and Lemma 2.7, which shows a point inside the convex hull of a set P can encode useful information, such as a 2-approximation of the diameter of P.
>
> > The authors explain the role of adaptive adversaries, motivating it. Such an adversary model is common in the literature, and the writing could be improved to focus more on the overview of technical challenges.
>
> There are different variations of adaptive adversaries in the literature. The model we consider in this paper is the strongest. We thought it would be good if we explained the differences between randomness-adaptive adversaries and the weaker version, the output-adaptive adversaries. In this way, it will be clear to the readers precisely what adversarial model we consider. However, you are right; the discussion is a bit long. In the updated version of our paper, we make this section more compact and expand the overview of our algorithm and the analysis.
>
>
> Finally, if you feel that our response has adequately addressed your major concerns, please, consider adjusting your score accordingly.

---

> > ### Author Response · Authors · 2025-08-03
> >
> > Dear Reviewer,
> >
> > We would like to kindly check whether you have any remaining questions or concerns. We would be happy to provide further clarification. If not, we would be grateful if you could consider updating your score in light of the rebuttal.

---

> ### Comment · Area_Chair_rXUG · 2025-08-04
> **Respond to author rebuttal**
>
> Dear Reviewer LwQb,
>
> Would you please check if the authors address your major concerns, and if you have further comments? A minimal response would be to submit your “Mandatory Acknowledgement”.
>
> Regards,
>
> AC

---

> > ### Comment · Reviewer_LwQb · 2025-08-04
> > **Rebuttal Response**
> >
> > I thank the authors for clarifying my questions. Please include the intuition and highlight the novelty in the updated version.

---

### Decision · Program_Chairs · 2025-09-17

**Decision:**

Accept (poster)

**Comment:**

The main result is a 2-approximate fully dynamic algorithm for diameter in high dimensional R^d, under adaptive adversary. A secondary result about k-center is also obtained.

The diameter/k-center in high dimension is a fundamental problem, and the adaptive adversary setting is also a popular topic. The ratio of 2 seems to be strong, albeit it is not justified if it is possible to do 1 + eps. The technique looks nontrivial to me. Other weaknesses include weaker bound for k-center (e.g., dependence on k and the ratio are large). The reviews are generally positive and most concerns are discussed and addressed during the rebuttal.

Overall, this is a solid paper that merits acceptance.